# Soil solution in Swiss forest stands: A 20 year's time series

**Sabine Braun** [ID] [1]*, **Simon Tresch** [ID] [1], **Sabine Augustin** [2]

**1** Institute for Applied Plant Biology, Witterswil, Switzerland, **2** Federal Office for the Environment, Berne, Switzerland

* sabine.braun@iap.ch

**Data Availability Statement:** https://doi.org/10.5061/dryad.2z34tmphm.

**Funding:** SB and ST: Federal Office for the Environment (measurements and manuscript writing), forest authorities of the cantons AG, BE, BL, BS, TG, SO, ZH and Environmental Offices of

## Abstract

Soil solution chemistry is influenced by atmospheric deposition of air pollutants, exchange processes with the soil matrix and soil-rhizosphere-plant interactions. In this study we present the results of the long-term Intercantonal Forest Observation Program in Switzerland with soil solution measurements since 1998 on a current total of 47 plots. The forest sites comprise two major forest types of Switzerland including a wide range of ecological gradients such as different nitrogen (N) deposition and soil conditions. The long-term data set of 20 years of soil solution measurements revealed an ongoing, but site-specific soil acidification. In strongly acidified soils (soil pH below 4.2), acidification indicators changed only slowly over the measured period, possibly due to high buffering capacity of the aluminum buffer (pH 4.2–3.8). In contrast, in less acidified sites we observed an increasing acidification rate over time, reflected, for example, by the continuous decrease in the ratio of base cations to aluminum (BC/Al ratio). Nowadays, the main driver of soil acidification is the high rate of N deposition, causing cation losses and hampering sustainable nutrient balances for tree nutrition. Mean nitrate leaching rates for the years 2005–2017 were 9.4 kg N ha$^{-1}$ yr$^{-1}$, ranging from 0.04 to 53 kg N ha$^{-1}$ yr$^{-1}$. Three plots with high N input had a remarkable low nitrate leaching. Both N deposition and nitrate leaching have decreased since 2000. However, the latter trend may be partly explained due to increased drought in recent years. Nonetheless, those high N depositions are still affecting the majority of the forest sites. Taken together, this study gives evidence of anthropogenic soil acidification in Swiss forest stands. The underlying long-term measurements of soil solution provides important information on nutrient leaching losses and the impact climate change effects such as droughts. Furthermore, this study improves the understanding of forest management and tree mortality regarding varying nitrate leaching rates.

## 1 Introduction

Since the 1980s it has been recognized that soil acidification due to anthropogenic input of sulfur and nitrogen compounds [1] poses a serious threat to forest health [2]. An increase in soil acidification, related to atmospheric acid deposition, has been reported in many European

Central Switzerland (long-term Intercantonal Forest Observation Program).

**Competing interests:** The authors have declared that no competing interests exist.

countries such as Germany [3], Sweden [4] or France [5]. The deposition of acidifying substances, in particular of sulfur compounds, has decreased in recent years in Europe, due to effective mitigation measures. In consequence, sulfate concentrations in soil solution have decreased significantly [6]. However, the development of nitrogen indicators is more divergent [7]. In Canada, for example, the chemical recovery of streams was slower than expected due to the reduction of acid deposition [8].

In Switzerland, acidification and eutrophication due to the deposition of reactive nitrogen compounds remain a critical issue [9],[10]. Between 1990 and 2017, emissions of sulfur compounds decreased by 85%, those of oxidized nitrogen by 55% and those of reduced nitrogen by 18% [11]. However, the input of nitrogen in forests in the proximity of intensive agriculture is still high with an average of 20.4 kg N ha$^{-1}$ yr$^{-1}$ and up to 50 kg N ha$^{-1}$ yr$^{-1}$ [12]. Exceedance of the critical loads for acidity and nitrogen is a continuing worldwide environmental problem, also in Switzerland [10]. For instance, Graf Pannatier et al. [14] observed a decrease in the BC/Al ratio in two out of five Swiss long-term forest monitoring sites between 1999 and 2007, which can be interpreted as an ongoing acidification during this time period.

Concerns about forest health led to the initiation of forest monitoring programs in the 1980s, where monitoring of soil solution is an important part [13]. The chemistry of soil solution is affected by atmospheric deposition, exchange processes between the solid and the dissolved phase in the soil and the nutrient uptake by the roots and other rhizosphere processes [14]. Stress indicators based on soil solution have been elaborated by expert groups under the International Cooperative Program on Modelling and Mapping of Critical Loads and Levels and Air Pollution Effects, Risks and Trends (ICP Modelling und Mapping) of the Geneva Air Convention (CLRTAP) of the UNECE [15]. Important chemical criteria for assessing the acidity of soil solution in forest ecosystems are the ratio between base cations (BC = $Ca^{2+}$ + $K^+$ + $Mg^{2+}$) and aluminum (BC/Al-ratio) [16], the pH value and the concentration of inorganic aluminum [15]. Aluminum bound in organic complexes is not toxic for plant roots [17]. In addition, critical thresholds have been identified for base saturation (BS) of the solid phase, for the alkalinity and for the acid neutralizing capacity of the soil solution [15].

Elevated nitrate leaching from the rooting zone in form of negatively charged nitrate induces acidification due to the concomitant loss of positively charged base cations from the soil. Acidification may impair root development [18], [19], [20] and nutrient supply to plants [3], [21], [22]. Sustainability calculations revealed that the loss of the cations $Ca^{2+}$, $Mg^{2+}$, and $K^+$ through nitrate leaching poses a greater risk to Swiss forest stands than cation exports with whole tree harvest [23]. To protect forests from such negative effects, maximum tolerable values for nitrate leaching have been defined [15]. Not only for forest ecology, but also for drinking water management, increased nitrate leaching is an important issue, as it poses a risk to human health [24].

The eutrophication effects of N have been addressed in another UNECE document [25]. Rihm and Achermann calculated the exceedance of the critical loads of N for Switzerland using a mass balance approach [10]. For productive forests they estimated an exceedance rate of 87% for the year 2015 [12]. Consequences of N excess have been highlighted by Aber [26] who developed a conceptual model for the effects of N saturation in forest ecosystems. This model describes different stages of the saturation process, with soil nitrate leaching occurring at a late stage, when N can no longer be used for growth. It was further developed by Aber [27], then confirmed and improved by Emmett [28]. Lovett & Goodale [29] suggested, based on results from an experiment in oak stands with very high N input, that many processes may occur simultaneously rather than in sequence. Observed N effects in forests differ according to site conditions such as climate, soil quality, and annual N input. In Scandinavia, Binkley & Högberg [30] concluded that increased N deposition led to higher forest growth without

causing quantifiable problems. These findings are in contrast to the overview on forest effects by Näsholm et al. [31] who listed numerous N impacts in Northern ecosystems. Data analysis from Swiss forests show that current forest growth is only slightly increased by N and that deposition rates >25 kg N ha$^{-1}$ yr$^{-1}$ show rather inhibitory effects [32], [35].

Nitrate leaching is strongly linked to atmospheric N deposition [33], [34]. The C:N of forest soil ratio is known to be a good predictor of the risk of nitrate leaching [35]. A strong increase of nitrate leaching was observed at C:N ratios < 25. At higher ratios nitrogen is immobilized and nitrification is inhibited [36]. In addition, high nitrate leaching rates were observed after strong disturbances such as tree cutting [37].

The aim of the present study is to analyze trends in soil solution data collected over a period of 20 years from currently 47 plots of the Intercantonal Forest Monitoring Program in Switzerland [38]. The observed changes in the element concentration of the soil solution measurements were analyzed with respect to international critical limits and other threshold values in order to assess the risk of acidification and eutrophication effects on forest health in Switzerland. The following research questions were discussed:

i.  Is there an exceedance of critical limits?

ii.  Do the reductions in acid depositions lead to corresponding changes in soil solution chemistry?

iii.  What are suitable predictors to recognize the risk of high nitrate leaching?

The parameters measured in this monitoring program are based on the Guidelines on Reporting Monitoring and Modelling of Air Pollution Effects of the Geneva Air Convention [39].

## 2 Materials and methods

### 2.1 Plots

The investigated sites are part of the Intercantonal Forest Observation Program in Switzerland [38]. Permission to perform the study was obtained from the local forest authorities and the forest owners. The first soil solution samplers were installed in nine out of 189 plots in 1997, and samples have been collected since 1998. In the following years, additional plots were included, resulting in a current total of 47 plots with soil solution measurements (Table 1, Fig 1). The plots cover a wide range of forest soils. Base saturation was determined using an unbuffered NH$_4$Cl extract [40]. The pH was measured in a suspension with 0.01 N CaCl$_2$ at a ratio of 1:2.5. A detailed description of the forest assessments carried out as well as other soil and foliar analyses is given by [32]. The rates of tree mortality and removed trees were derived from annual observations and combined into one variable referred to as "tree removal rates". This variable was further divided into the four following lagged effects: the tree removal rate of the current year (lag0), the previous year (lag1), the last two years (lag2) and the last three years (lag3).

### 2.2 Soil solution

For each site and soil depth, eight soil solution samplers (ceramic suction cups, 0653X01-B0.5M2, Soilmoisture Equipment Corp.) were installed in the topsoil and five in the subsoil. Actual depths varied according to soil condition but a frequent sampling design was 20, 50 and 80 cm. Detailed site-specific information and time series can be found in [43]. Following the monthly sampling of the soil solution, the samples from the same location and from the same depth were pooled. We measured pH (Metrohm pH-meters 716 and 809, with

**Table 1. Site properties of forest plots with soil solution samplers. pH: pH(CaCl$_2$) in the uppermost 40 cm of the soil, BS: Base saturation in the uppermost 40 cm of the soil (%).** CN: C:N ratio in the forest floor or the uppermost humus horizon. Prec: precipitation in mm, average 1981–2018. Leaching: leaching water flux in mm, calculated with the hydrological model Wasim-ETH [41], average 1981–2018. Species: tree species: Fa beech, Pic Norway spruce, Ab fir, La larch, Pin pine. Soil types: FAO classification. Weath: weathering rate in keq ha$^{-1}$ a$^{-1}$: calculations with SAFE [42] for the rooting zone (0–60 cm). Start: starting year of the soil solution measurements. Site Muri (storm) was cleared in 1999 during the gale "Lothar".

| Site | abbr. | altitude (m) | prec (mm) | leaching (mm) | species | pH | BS (%) | CN | soil type | weath keq ha$^{-1}$ yr$^{-1}$ | start (year) |
|---|---|---|---|---|---|---|---|---|---|---|---|
| Aarwangen | AW | 470 | 1140 | 482 | Fa | 3.99 | 10 | 14.5 | Dystric Cambisol | 1.31 | 2002 |
| Aeschau | AU | 940 | 1512 | 783 | Ab Pic (Fa) | 3.67 | 20 | 26.0 | Dystric Arenosol | 0.45 | 1997 |
| Aeschi | AI | 510 | 1160 | 472 | Fa Pic | 3.87 | 15 | 21.2 | Haplic Luvisol | 1.24 | 1998 |
| Allschwil | AL | 350 | 896 | 153 | Pic | 4.31 | 88 | 14.0 | Haplic Luvisol | | 2006 |
| Bachtel | BAB | 1030 | 1825 | 1093 | Fa | 3.93 | 36 | 15.6 | Chromic Luvisol | 4.62 | 1999 |
| Bachtel | BA | 1040 | 1770 | 998 | Pic | 4.01 | 7 | 24.8 | Chromic Luvisol | 0.90 | 1997 |
| Beromünster | BE | 640 | 1220 | 321 | Pic | 5.00 | 90 | 23.1 | Gleyic Cambisol | 7.06 | 2016 |
| Bonfol | BO | 450 | 1091 | 417 | Fa | 4.26 | 18 | 20.3 | Dystric Cambisol | 0.67 | 2004 |
| Braunau | BRAU | 710 | 1253 | 400 | Pic | 4.05 | 55 | 19.8 | Haplic Luvisol | | 2006 |
| Breitenbach | BB | 460 | 1111 | 346 | Fa | 4.53 | 91 | 14.3 | Haplic Luvisol | 0.85 | 2003 |
| Brislach beech | BRB | 435 | 1041 | 378 | Fa | 4.09 | 25 | 13.3 | Haplic Luvisol | 0.88 | 2000 |
| Brislach spruce | BR | 435 | 1042 | 258 | Pic | 3.93 | 12 | 23.3 | Haplic Luvisol | 0.84 | 1997 |
| Bürglen | BUR | 640 | 1582 | 572 | Pic | 4.77 | 99 | 22.2 | Cambisol | 0.36 | 2016 |
| Busswil | BU | 600 | 1195 | 388 | Pic | 3.78 | 3 | 18.9 | Haplic Luvisol | 0.99 | 2004 |
| Diessenhofen | DI | 520 | 942 | 290 | Pic | 3.77 | 16 | 20.8 | Dystric Cambisol | | 2006 |
| Frienisberg | FR | 725 | 1209 | 542 | Fa Pic | 3.90 | 21 | 21.2 | Dystric Arenosol | 0.64 | 1997 |
| Gelfingen | GE | 540 | 1135 | 451 | Fa | 6.55 | 100 | 21.9 | Calcaric Cambisol | 1.59 | 2016 |
| Giswil | GI | 540 | 1306 | 479 | Fa | 5.86 | 100 | 19.5 | Calcaric Cambisol | 10.84 | 2016 |
| Grenchenberg | GB | 1220 | 1511 | 961 | Fa Pic | 5.64 | 100 | 15.1 | Calcaric Cambisol | 19.05 | 1997 |
| Grosswangen | GW | 600 | 1114 | 320 | Pic | 3.52 | 14 | 21.9 | Stagnic Acrisol | 1.25 | 2016 |
| Habsburg K | HA | 430 | 1072 | 308 | Fa | 4.17 | 16 | 17.1 | Haplic Luvisol | 0.84 | 2004 |
| Hinwil | HI | 650 | 1456 | 619 | Pic | 5.12 | 95 | 15.4 | Eutric Cambisol | 1.33 | 2002 |
| Le Châtelard | LC | 1010 | 1654 | 811 | Pic | 3.74 | 20 | 29.3 | Gleyic Cambisol | 1.53 | 2006 |
| Lurengo spruce | LUB | 1620 | 1786 | 1098 | Pic Pin La | 3.90 | 28 | 26.2 | Dystric Arenosol | | 1999 |
| Lurengo N exp. | LU | 1600 | 1786 | 1123 | Pic La | 4.17 | 19 | 22.5 | Podzol | 0.59 | 1997 |
| Möhlin | MO | 290 | 1034 | 267 | Pic | 3.79 | 12 | 17.5 | Haplic Luvisol | 1.22 | 1998 |
| Muri beech | MUB | 490 | 1110 | 340 | Fa | 4.00 | 24 | 18.3 | Haplic Luvisol | 0.56 | 1999 |
| Muri spruce | MUF | 490 | 1104 | 278 | Pic | 3.88 | 10 | 26.5 | Dystric Cambisol | 0.74 | 2001 |
| Muri storm | MU | 490 | 1104 | 588 | Pic | 4.08 | 23 | 18.9 | Haplic Luvisol | 0.62 | 1997 |
| Muttenz | MUU | 375 | 912 | 228 | Fa | 4.06 | 41 | 15.7 | Stagnic Luvisol | 0.50 | 2004 |
| Oberschrot | OS | 950 | 1340 | 541 | Fa Pic | 3.61 | 11 | 17.2 | Gleyic Stagnic Cambisol | | 2006 |
| Olsberg | OL | 380 | 998 | 240 | Fa | 4.06 | 20 | 15.4 | Dystric Planosol | 0.48 | 2004 |
| Pratteln | PR | 415 | 966 | 339 | Fa | 5.15 | 100 | 12.4 | Chromic Luvisol | 2.24 | 2002 |
| Rafz | RAF | 540 | 995 | 315 | Pic | 4.18 | 16 | 19.0 | Haplic Luvisol | 0.70 | 2004 |
| Riehen | RI | 470 | 1005 | 402 | Fa | 6.41 | 100 | 13.3 | Haplic Luvisol | 1.26 | 2002 |
| Rünenberg | RU | 590 | 1017 | 245 | Fa | 4.13 | 35 | 17.2 | Haplic Luvisol | 0.68 | 2002 |
| Sagno | SA | 770 | 1782 | 943 | Pic | 3.83 | 25 | 21.8 | Eutric Cambisol | 0.41 | 1999 |
| Scheidwald | SW | 1170 | 1500 | 547 | Pic | 3.41 | 7 | 27.9 | Dystric Gleysol | 0.66 | 2008 |
| Sempach | SE | 550 | 1139 | 450 | Fa | 3.71 | 39 | 21.6 | Gleyic Luvisol | 2.32 | 2016 |
| Stans | ST | 560 | 1437 | 924 | Fa | 6.40 | 100 | 17.4 | Calcaric Cambisol | 28.30 | 2016 |
| Wangen | WG | 500 | 1143 | 450 | Fa Pic | 3.88 | 24 | 23.3 | Chromic Luvisol | | 2008 |
| Wangen SZ | WSZ | 470 | 1536 | 634 | Fa | 4.43 | 95 | 14.8 | Luvisol | 1.77 | 2016 |
| Wengernalp | WA | 1870 | 1605 | 922 | Pic | 3.53 | 28 | 14.2 | Podzol | 0.15 | 1997 |

*(Continued)*

**Table 1.** (Continued)

| Site | abbr. | altitude (m) | prec (mm) | leaching (mm) | species | pH | BS (%) | CN | soil type | weath keq ha$^{-1}$ yr$^{-1}$ | start (year) |
|---|---|---|---|---|---|---|---|---|---|---|---|
| Winterthur | WI | 530 | 1178 | 465 | Pic | 5.25 | 97 | 16.0 | Vertisol | 18.46 | 2003 |
| Zofingen | ZO | 540 | 1130 | 370 | Fa Pic | 4.00 | 17 | 17.9 | Haplic Luvisol | 0.92 | 2004 |
| Zugerberg HG | ZBB | 980 | 1569 | 900 | Fa Pic | 4.20 | 37 | 19.8 | Eutric Cambisol | 0.63 | 1999 |
| Zugerberg (N-exp.) | ZB | 940 | 1457 | 821 | Fa | 3.91 | 100 | 18.5 | Dystric Cambisol | 0.45 | 1997 |
| Zugerberg VG | ZV | 900 | 1457 | 550 | Pic | 3.62 | 24 | 20.2 | Dystric Cambisol | 1 | 2002 |

Metrohm Aquatrode), conductivity (Metrohm conductivity meters 712 and 856, with Metrohm cell 6.0916.040) and alkalinity (titration with HCl to a pH of 4.35 (Metrohm 809)) of the soil solution immediately after sampling. Samples were then kept frozen (-20˚C) until further analysis. For cation analysis the samples were acidified with 0.5 ml $HNO_3$ in 10 ml solution prior to freezing. For anion analysis the samples were filtered through a 0.45 μm membrane filter. $Ca^{2+}$, $Mg^{2+}$, $Al^{3+}$ and $Mn^{2+}$ were analyzed using atomic absorption photometry (Varian 640) and $K^+$, $Na^+$ by flame photometry (Varian 640). Inorganic Al was measured as difference before and after passing the samples through an ion exchanger (0.5 ml IC-H, Alltech 30264). $NH_4^+$ was measured by photometric determination with indophenol blue [44].

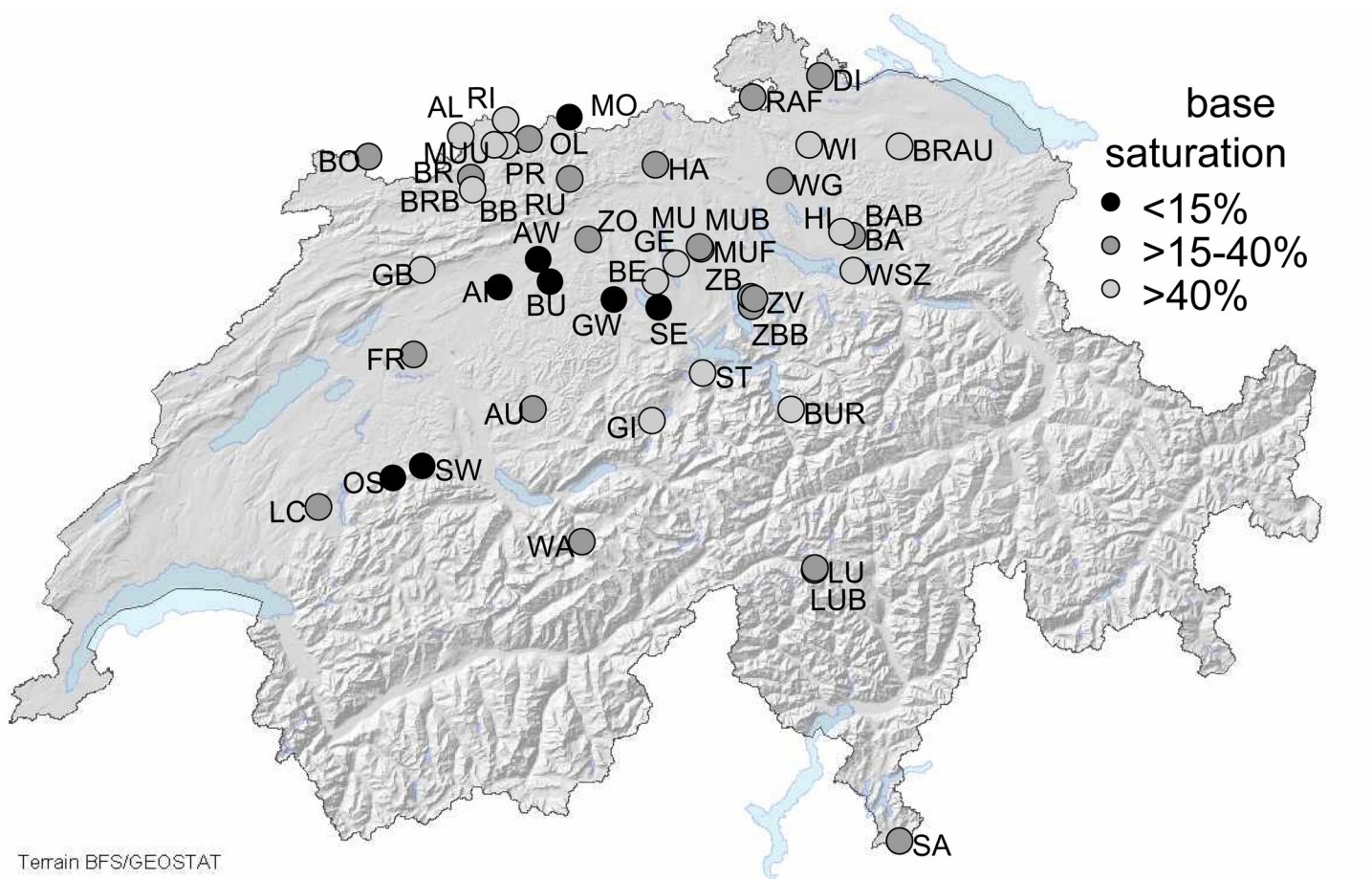

**Fig 1. Forest plots with soil solution samplers (sampled in 2017 and 2018), grouped according to the base saturation of the topsoil (average 0–40 cm).**

$NO_3^-$, $SO_4^{2-}$ and $Cl^-$ were assessed by ion chromatography with suppressed conductivity (Dionex GP50 pump, ED50 electrochemical detector and AS3500 autosampler). Dissolved organic carbon was measured by UV absorption at 280 nm according to [45].

Quality control was achieved by calculation of the ion balance, by comparison of measured and calculated conductivity [46,47] and by analysis of reference samples distributed once a year by the Norwegian Institute for Air Research (NILU).

The relation between base cations and aluminum (BC/Al) was calculated on a molar basis [15], using the concentration of inorganic aluminum. The determination of organic aluminum started in 2005. In order to get a homogenous time series for older data, the average proportion of organic aluminum to total aluminum was calculated for each soil layer. This proportion was then applied to data from 1998 to 2005. It varied between 50% for the uppermost soil water samplers and 25% in the lowest ones (S3 Fig in S1 File).

The amount of leaching water in mm was calculated using the hydrological model Wasim-ETH [41] taking into account soil characteristics (pF curve, texture), current vegetation cover and daily meteorological data interpolated for each site [32]. Leaching fluxes calculated for each sampling period were multiplied with concentrations to calculate element fluxes.

**2.2.1 Critical limits in soil solution.** The molar ratio between base cations (BC = $Ca^{2+}$, $Mg^{2+}$, $K^+$) to aluminum ($Al^{3+}$), the BC/Al ratio, in the soil solution is an important criterion for evaluating soil acidification. It has been shown to be closely correlated with growth and vitality parameters of the vegetation [16]. Initially, a limit value of 1 has been set for the BC/Al ratio [48], [16]. Since this limit value is not necessarily sufficient to protect forests, Ouimet et al. [49] suggested a BC/Al limit value of 10 for calculations of critical loads in Canada. Based on these findings, the critical limits of the BC/Al ratio were revised [15]. In Switzerland a revised BC/Al limit value of 7 has recently been applied [50].

Other critical limit values for the soil solution are a pH of 4 and a Al concentration of 0.2 eq m$^{-3}$ [15]. The Acid Neutralizing Capacity (ANC), an indicator of vulnerability to acidification, is defined as sum of the base cations $Ca^{2+}$, $Mg^{2+}$, $K^+$ and $Na^+$ minus the sum of the anions nitrate, sulfate and chloride. It relates the two criteria Al and proton concentration, given a $pK_{Gibbsit}$ of 8.04 [15]:

$$ANC = -Al - H$$

By solving this equation for $Al_{crit}$ of 0.2 eq m$^{-3}$ and pH 4, the maximum allowable leaching of alkalinity from the rooting zone is -300 µeq l$^{-1}$ [51].

While the above mentioned acidity indicators refer to soil solution, an indicator widely used in forestry is the base saturation (the cations $Ca^{2+}$, $Mg^{2+}$, $K^+$ on the cation exchange complex in BS) of the soil solid phase [52]. The parameter BS reflect the potential supply of cations to the trees. The relation between the cations in the soil solution and the BS can be described with the exchange equations according to Gapon or Gaines-Thomas; [53]. This relation between the soil solid phase and the soil solution has been examined, among others, by Hildebrand [54] and Schall et al. [55] who both stated that the exchange relationships depend on the chemical status of the soil. In the present study, we provide field data of the relation between the exchange complex and soil solution sampled in situ.

Monthly soil solution data were compared with the critical limits listed above. In order to avoid sensitivity to single outliers, an exceedance was indicated when more than 1% of the values were above the limit value.

Eutrophication effects of N input can be evaluated by using the concentrations and total amounts of N-leaching. Critical limits have been set accordingly. A concentration of >0.2 mg N l$^{-1}$ in soil solution has been related to changes in ground vegetation and in tree nutrition

[15]. For temperate deciduous forests, a threshold for a total annual nitrate leaching amount of 2–4 kg N ha$^{-1}$ yr$^{-1}$ has been set to avoid excessive base cation leaching and acidification. Absolute limits for nitrate leaching, based on concentrations in the soil solution, are particularly important for areas with high precipitation. Under such conditions, high losses of cation nutrients and thus reduced base saturation can occur as a consequence.

## 2.3 Weathering rates

Weathering rates were calculated using the model SAFE [42] for a subset of monitoring plots. The calculations are based on the measured mineralogy of soil samples [56]. The results were summed up for each layer down to a depth of 60 cm, taking into account deeper depths where dense rooting occurred at greater depths. This differentiation has been based on correlation analysis between soil chemistry and foliar analysis suggesting that nutrient uptake is considerably small below 60 cm [23]. Recent analyses of Al concentration in tree rings suggest that the SAFE model gives reasonable estimates of base saturation and thus also of weathering rates (Hopf et al., unpublished results).

## 2.4 Atmospheric deposition

Data on atmospheric deposition of reactive nitrogen compounds and base cations were received from the Swiss Federal Office for the Environment [12]. N deposition was modelled in a spatial resolution of 0.1 ha and the deposition of base cations with a spatial resolution of 2 km [57]. Model comparisons of N deposition with observational data showed a strong agreement [58], with the exception of the plots in Southern Switzerland, where the import of N compounds by air from Italy were more difficult to take into account. The deposition of base cations in Southern Switzerland was modelled according to [59].

## 2.5 Statistics

In order to test if the BC/Al ratio depends on the degree of acidification, a moving time window of 5 years were formed for the dependent variable and pH in soil solution. The development within these time windows was analyzed using a linear mixed effect model with plot as random effect (R, package lme4 [60]. Including the BC/Al ratio and soil solution pH at the start of the 5 year period, base saturation of the soil in the corresponding soil horizon (%), weathering rate of base cations (keq ha-1 yr-1), proportion of coniferous trees in the plot (%), modelled N deposition (kg N ha 1 yr 1), organic carbon in the corresponding soil horizon (% C), C:N and N:P ratio in the forest floor, clay content (%) and soil depth (binary variable coded as ≤70 (0) and >70 cm (1)).

For the relation between BC/Al in soil solution and the chemistry of the solid phase the properties of the horizon at the depth of the soil solution samplers were used as described above.

Explanatory variables for nitrate leaching were analyzed based on annual means. Linear mixed effect models with plot and year as random effects were used including modelled N deposition (kg N ha$^{-1}$ yr$^{-1}$), C:N ratio in the uppermost soil horizon, water holding capacity (0–100 cm in mm), annual minimum of site water balance (mm), rate of seepage water (mm), tree removal rate (current year and 3 lagged effects), shrub cover of the plot from a vegetation survey, proportion of coniferous trees in the plot and altitude (m).

Predictors were selected backwards using the Akaike Information Criterion (AIC). When the number of predictors had to be reduced to avoid oversaturation of the model, the Bayes Information Criterion (BIC) criterion was used. Residuals were examined for normal distribution, homoskedasticity and outliers using diagnostic plots. In the case of BC/Al and nitrate leaching a log transformation was required. Regression plots were produced using the R

functions ggpredict [61] and ggplot [62]. The former extracts predictions including 95% confidence intervals from a multivariate model taking the mean value of all other predictors. Pseudo-$R^2$ for mixed regression models were calculated according to Nakagawa and Schielzeth [63]. All R-codes and data for the models including diagnostic and effect plots are provided in the supplementary materials.

## 3 Results

### 3.1 Data description and time trends

There is a significant time trend for the measured indicators of acidity pH, ANC and BC/Al ratio as well as for the cations Ca, Mg, K, Al and the anions $NO_3^-$ and $SO_4^{2-}$, taken together mean concentrations per element and depth in all plots (Table 2). The only exceptions are

**Table 2. Summary statistics for the dataset by depth layer. Mean and confidence interval: estimates corrected for the varying dataset (mixed regression).** Minimum and maximum: median values of single years per site and depth. Regression against time: time trend with mixed regression of log transformed predictors (except pH and ANC). n = 20211 monthly samples, 47 plots, 22 years.

| element | unit | depth | Statistics | | | 95%-Confidence interval | | time trend | | |
|---|---|---|---|---|---|---|---|---|---|---|
| | | (cm) | mean | min | max | low | high | coeff. | se | p value |
| Aciditiy indicators | | | | | | | | | | |
| pH | | <30 | 5.24 | 4.04 | 8.03 | 4.96 | 5.51 | 0.006 | 0.001 | <0.001 |
| pH | | 30–60 | 5.52 | 4.10 | 8.33 | 5.16 | 5.89 | 0.017 | 0.001 | <0.001 |
| pH | | >60 | 6.05 | 4.11 | 8.43 | 5.74 | 6.36 | 0.027 | 0.001 | <0.001 |
| ANC | µeq l$^{-1}$ | <30 | 98 | -445 | 2620 | -31 | 227 | 0.903 | 0.556 | 0.104 |
| ANC | µeq l$^{-1}$ | 30–60 | 253 | -606 | 4520 | -3 | 508 | 1.369 | 0.580 | 0.018 |
| ANC | µeq l$^{-1}$ | >60 | 98 | -601 | 5744 | -31 | 227 | 3.053 | 1.027 | 0.003 |
| BC/Al | | <30 | 3.20 | 0.72 | >10000 | 2.47 | 4.17 | -0.020 | 0.001 | <0.001 |
| BC/Al | | 30–60 | 3.71 | 0.75 | >10000 | 2.63 | 5.22 | -0.014 | 0.001 | <0.001 |
| BC/Al | | >60 | 6.35 | 1.22 | >10000 | 4.79 | 8.44 | -0.015 | 0.001 | <0.001 |
| Cations | | | | | | | | | | |
| Ca | mg l$^{-1}$ | <30 | 2.28 | 0.08 | 43.04 | 1.52 | 3.42 | -0.030 | 0.002 | <0.001 |
| Ca | mg l$^{-1}$ | 30–60 | 2.43 | 0.04 | 105.2 | 1.39 | 4.28 | -0.030 | 0.002 | <0.001 |
| Ca | mg l$^{-1}$ | >60 | 4.15 | 0.11 | 100.7 | 2.57 | 6.71 | -0.016 | 0.001 | <0.001 |
| Mg | mg l$^{-1}$ | <30 | 0.54 | 0.07 | 8.32 | 0.41 | 0.73 | -0.022 | 0.001 | <0.001 |
| Mg | mg l$^{-1}$ | 30–60 | 0.58 | 0.08 | 12.26 | 0.41 | 0.81 | -0.028 | 0.001 | <0.001 |
| Mg | mg l$^{-1}$ | >60 | 0.89 | 0.08 | 17.58 | 0.64 | 1.25 | -0.021 | 0.001 | <0.001 |
| K | mg l$^{-1}$ | <30 | 0.21 | 0.01 | 5.12 | 0.16 | 0.30 | -0.036 | 0.002 | <0.001 |
| K | mg l$^{-1}$ | 30–60 | 0.18 | 0.01 | 1.59 | 0.13 | 0.25 | -0.047 | 0.002 | <0.001 |
| K | mg l$^{-1}$ | >60 | 0.14 | 0.01 | 2.06 | 0.11 | 0.18 | -0.033 | 0.001 | <0.001 |
| Al | mg l$^{-1}$ | <30 | 0.19 | 0.01 | 3.77 | 0.13 | 0.29 | 0.018 | 0.002 | <0.001 |
| Al | mg l$^{-1}$ | 30–60 | 0.14 | 0.01 | 3.80 | 0.09 | 0.24 | 0.003 | 0.002 | 0.222 |
| Al | mg l$^{-1}$ | >60 | 0.07 | 0.01 | 3.61 | 0.05 | 0.10 | 0.016 | 0.002 | <0.001 |
| Anions | | | | | | | | | | |
| $NO_3^-$ | mg N l$^{-1}$ | <30 | 0.66 | 0.01 | 11.83 | 0.37 | 1.16 | -0.049 | 0.003 | <0.001 |
| NO3 | mg N l$^{-1}$ | 30–60 | 0.52 | 0.01 | 77.39 | 0.25 | 1.06 | -0.025 | 0.003 | <0.001 |
| NO3 | mg N l$^{-1}$ | >60 | 0.20 | 0.01 | 19.55 | 0.11 | 0.38 | -0.055 | 0.003 | <0.001 |
| $SO_4^{2-}$ | mg S l$^{-1}$ | <30 | 0.79 | 0.06 | 6.46 | 0.61 | 1.03 | -0.050 | 0.001 | <0.001 |
| $SO_4^{2-}$ | mg S l$^{-1}$ | 30–60 | 1.12 | 0.09 | 11.25 | 0.82 | 1.54 | -0.047 | 0.001 | <0.001 |
| $SO_4^{2-}$ | mg S l$^{-1}$ | >60 | 1.73 | 0.14 | 23.18 | 1.36 | 2.21 | -0.038 | 0.001 | <0.001 |

ANC in <30 cm depth and Al in 30–60 cm depth. The pH has generally increased with time while the BC/Al ratio has decreased. These two parameters give thus conflicting interpretation. However, the base cations have all decreased while Al concentrations have increased.

## 3.2 Acidification

**3.2.1 Acidification status, comparison with thresholds.** The exceedance of acidity limits according to the Geneva Air Convention [15] or suggested by [64] are listed in Table 3. A BC/Al ratio lower than 1 in at least one layer was observed in 27% of the plots and a BC/Al ratio lower than 7 was observed in 71% of the plots. The frequency distribution of the BC/Al ratio and ANC are shown in S1 Fig in S1 File.

The BC/Al ratio in soil solution was regressed against base saturation in the corresponding soil layer (Fig 2A) and against pH(CaCl$_2$) of the solid phase (Fig 2B). The relation with base saturation is stronger (p = <0.001, Adj.R$^2$ = 0.73, S1 Table in S1 File) than with pH(CaCl$_2$) (p = <0.01, Adj.R$^2$ = 0.55, S1 Table in S1 File). This difference can be attributed to the high buffer capacity of the aluminum buffer (pH 3.8 –pH 4.2; 150 kmol H$^+$ per % clay), reflected by the cluster of points around pH 4 (Fig 2B). For BS = 20 the regression function predicts a BC/Al ratio of 10 (95% CI 7.7–12.6); for BS = 40 a BC/Al of 28 (22.4–33.9). These relationships allow–within certain limits–to link BC/Al ratios with base saturation values which are more often used as acidity indicator in forestry.

**3.2.2 Leaching of base cations.** The leaching of base cations is relevant for the assessment of acidification and the evaluation of the sustainability of forest nutrition. The relation between the Ca input from weathering and Ca leaching was clearly significant (p = <0.001, Adj.R$^2$ = 0.52). In 35 out of 41 plots (85%), Ca leaching exceeded the Ca input by weathering (Fig 3A). When atmospheric deposition of Ca is added to weathering (Fig 3B), 16 plots (39%) still have a negative Ca balance. The highest Ca losses were observed in weakly buffered sites in the silicate and exchange buffer ranges (pH 4.2–6.2; WI, HI, PR, BB) as well as on one site with very high N input (SA). Leaching of Ca was correlated significantly with weathering rate (p = <0.001, Adj.R$^2$ = 0.59) and soil water holding capacity (p = <0.01), while neither N deposition nor species composition of the tree layer were significant predictors (S6 Table in S1 File). The same analysis with leaching rates at different time periods slightly reduced the proportion of plots with a negative balance at the later time period. For instance, between 2015–2018, Ca leaching exceeded the Ca weathering input in 83% of the plots.

**3.2.3 Development of acidification.** The BC/Al ratio decreased significantly over time in all depth and base saturation levels (S7 Table in S1 File). Even in base rich soils there was a clear decrease of the BC/Al ratio, i.e. a clear increase in acidification (Fig 4). The decrease of BC/Al ratio over time was strongest in the uppermost soil depth (S7 Table in S1 File). Below the rooting zone this decrease was weaker but still significant.

**Table 3. Frequency of plots with exceedance of various acidity limits in the years 2013–2018.** Number of sites = 47.

| Acidity limit | Reference | Frequency of plots with exceedance (%) |
|---|---|---|
| BC/Al <1 | [15] | 27 |
| BC/Al<7 | [15] | 71 |
| ANC < -500 µeq l$^{-1}$ | [64] | 22 |
| ANC <-300 µeq l$^{-1}$ | [15] | 37 |
| ANC <0 µeq l$^{-1}$ | [64] | 86 |
| pH <4 | [15] | 8 |
| Al >0.2 eq m$^{-3}$ | [15] | 0 |

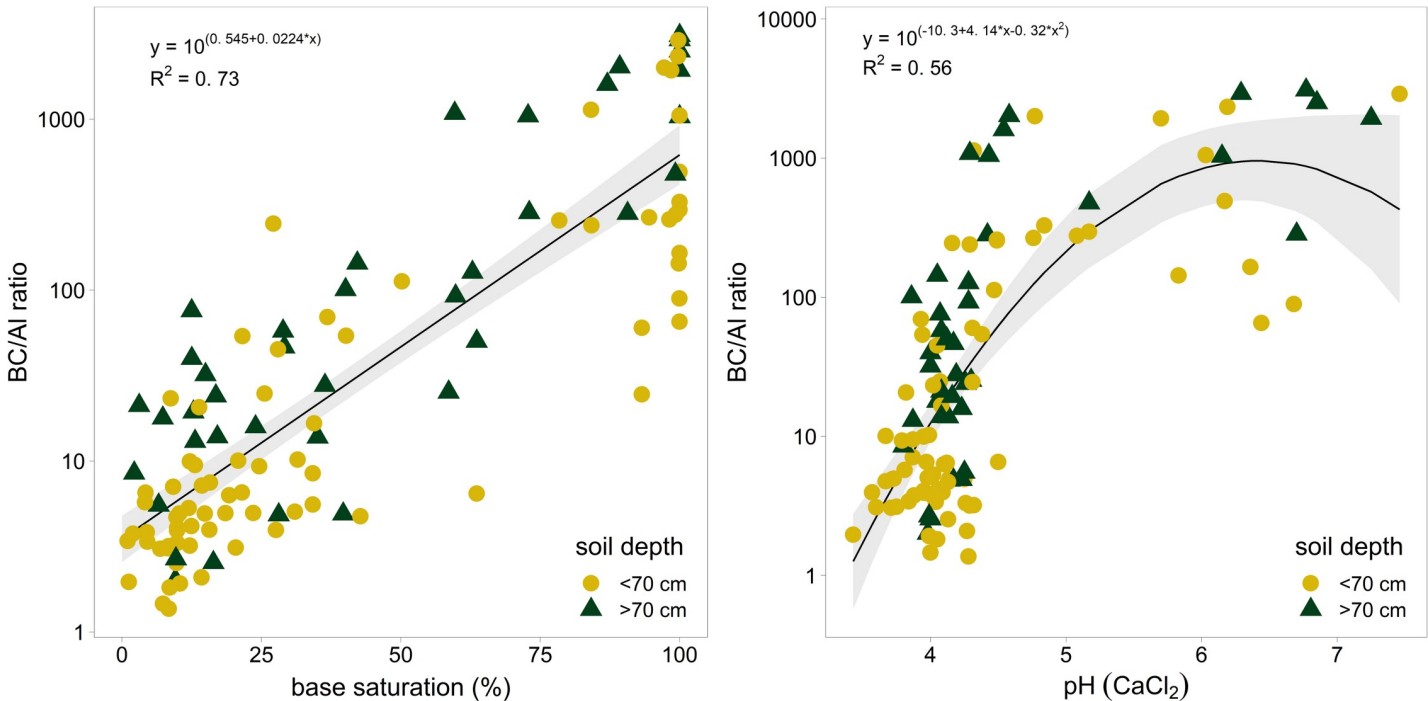

**Fig 2.** Relationship between the BC/Al ratio of the soil solution and base saturation (A) and between the BC/Al ratio and soil pH (B). Points represents measurements from the topsoil (depth 0–70 cm), triangles from the subsoil (>70 cm depth). Model outputs with the corresponding coefficients are given in S1 Table in S1 File.

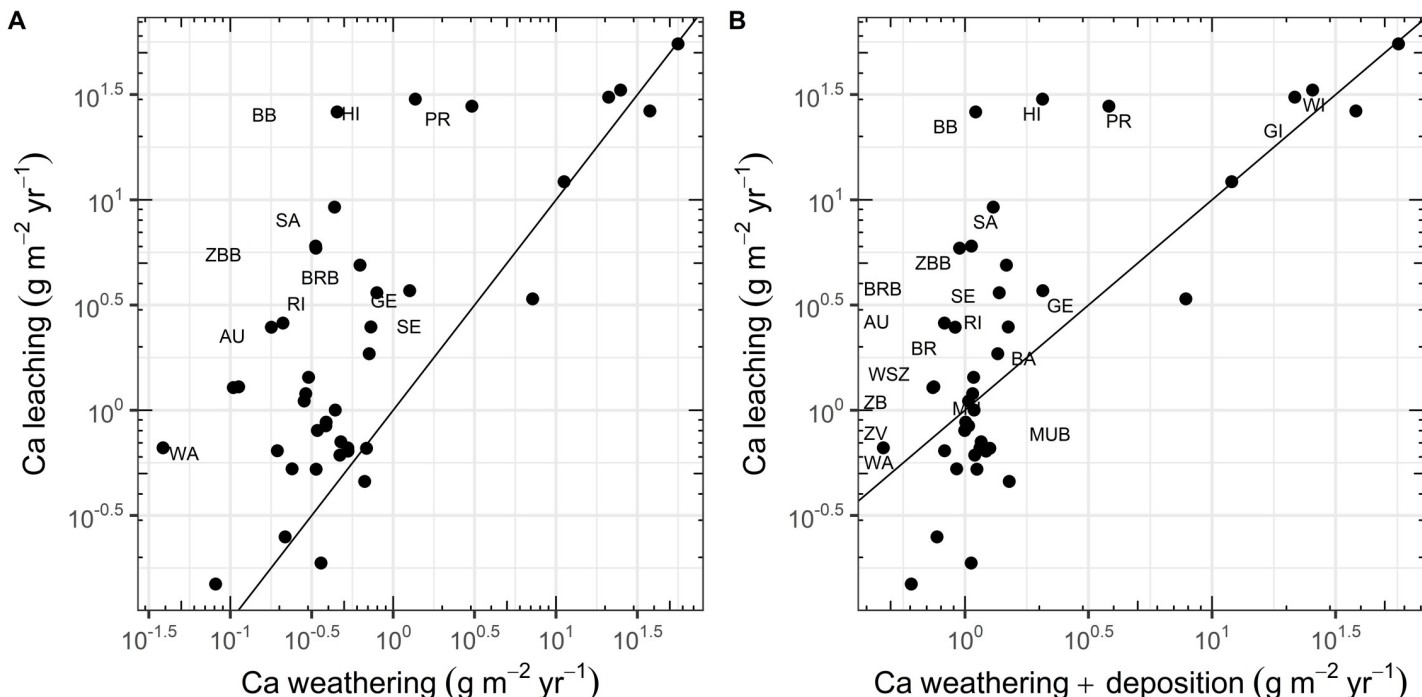

**Fig 3. Relation between the leaching of Ca at a depth of 60–80 cm (average 2005–2018) and the weathering rate of base cations cumulated over the uppermost 60 cm.** A: Relation between Ca leaching and weathering rate. B: Relation between Ca leaching and the sum of Ca weathering and Ca deposition. Site abbreviations can be found in Table 1. Only sites with Ca leaching exceeding the weathering input by a factor of 1.1 are labelled in the plot. The line represents the 1:1 line.

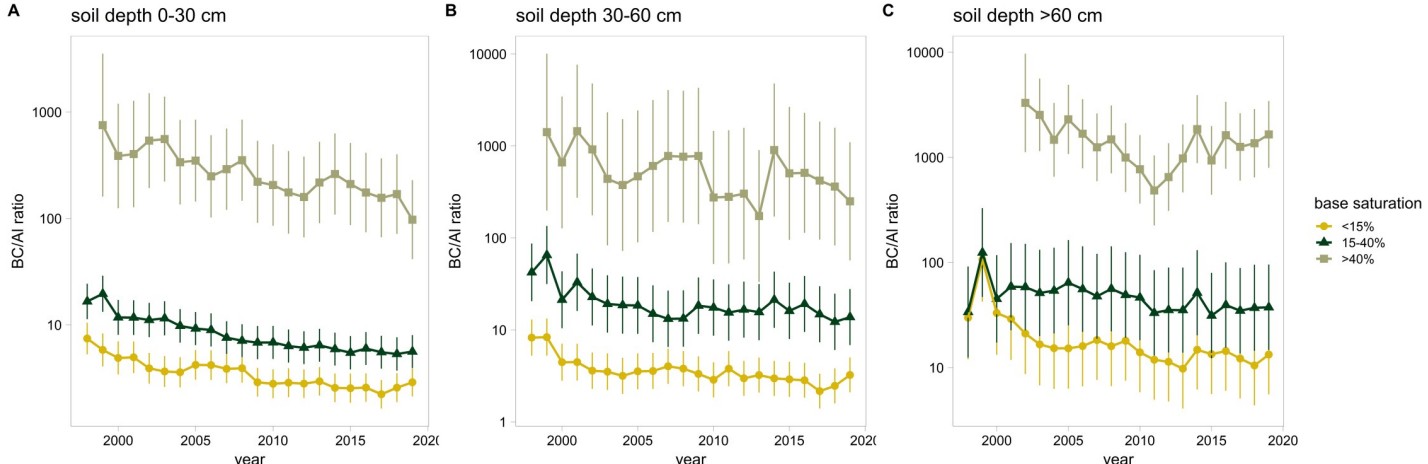

**Fig 4. Development of the BC/Al ratio over time in soils of different soil depths and base saturation levels.** Error bars: 95% confidence interval, extracted from the mixed regression models (S7 Table in S1 File).

The declining trends became weaker the more acidified the soil was. To examine this association, the changes of BC/Al ratio within five years were analyzed in relation to the initial status of the sample in a moving window analysis. The resulting regression (Table 4, Fig 5) supports the hypothesis that the trends are related to acidification status. Significant predictors for the trend were the initial BC/Al ratio, the initial pH, BS and soil depth. Predicted changes of log BC/Al in Fig 5 below zero correspond to an expected decrease of the BC/Al ratio during the following five years. This means that a decrease of BC/Al is expected when the BS is below 48%, the pH below 5.9 or the BC/Al ratio above 16.3. The pH value of 5.9 corresponds well with the lower limit of the Ca buffer range (pH 6.2), according to the equilibrium of $CaCO_3$ with $H_2CO_3$ in soil [65], [66]. These values may be regarded as "thresholds" for acidification under the current deposition situation. No interpretation was found for the unexplained part of this regression, addressed as residuals.

**3.2.4 Nitrate leaching.** Mean annual nitrogen leaching rate was 9.4 kg N ha$^{-1}$ yr$^{-1}$ for the years 2005–2018 (S2 Table in S1 File). Nitrogen leaching decreased significantly between 1998 and 2018 (Fig 6). The proportion of plots exceeding the leaching limits of the Geneva Air Convention [15] decreased from 83% (1998, 12 plots) to 34% (2018, 47 plots). The two plots with the highest average leaching rates (S2 Table in S1 File) have very different properties. Plot SA in Southern Switzerland has a leaching rate of 52 kg N ha$^{-1}$ yr$^{-1}$, resulting from a high N deposition, leading to an average nitrogen concentration of 6.7 mg N l$^{-1}$, and a high output with the seepage water (>900 mm yr$^{-1}$). Whereas, the second plot AL in Northwestern Switzerland is characterized by extremely high N concentrations in soil water (average 30 mg N l$^{-1}$) and a low

**Table 4. Changes of BC/Al ratio within five years and various parameters.** Dependent variable: Difference in log transformed BC/Al during five years. Pseudo-$R^2$ fixed effects = 0.37, Pseudo-$R^2$ including random effect = 0.53, n = 994).

|  | coeff | SE | p value |
| --- | --- | --- | --- |
| (Intercept) | -0.218 | 0.024 | <0.001 |
| Initial BC/Al-ratio | -0.132 | 0.006 | <0.001 |
| Initial soil solution pH | 0.057 | 0.005 | <0.001 |
| base saturation (%) | 0.0014 | 0.002 | <0.001 |
| depth coded (0: <70 cm, 1:> = 70 cm) | 0.046 | 0.005 | <0.001 |

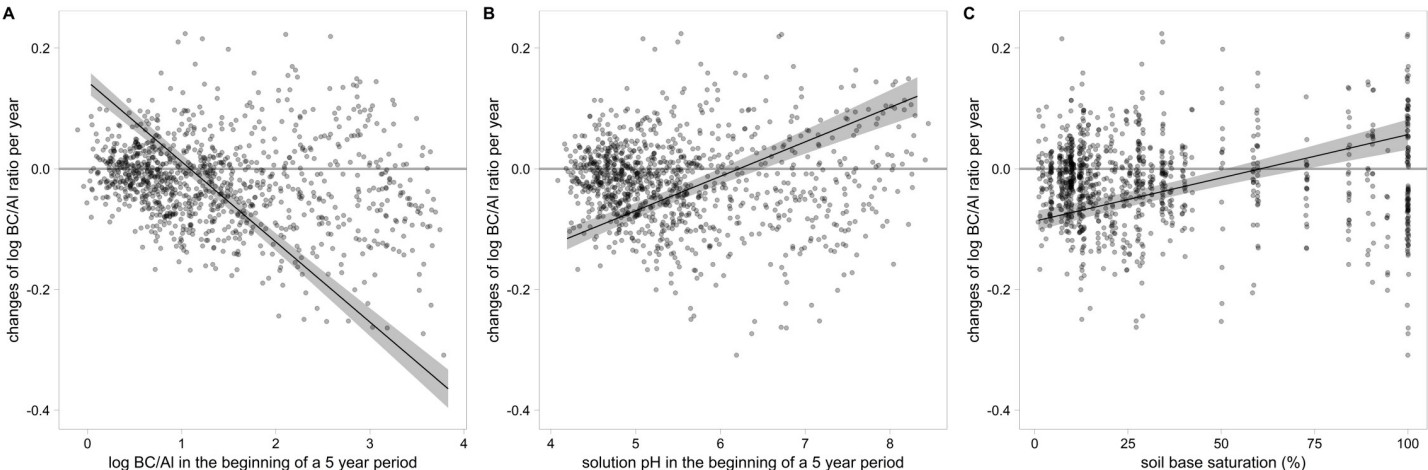

**Fig 5. Rate of changes of BC/Al ratio within five years in relation to selected predictors from Table 4.** Predicted values including 95% confidence intervals are conditioned on all other fixed effects. Negative changes signify an expected decrease in BC/Al. The linearity of the relations was tested using polynomial functions.

leaching water flux, resulting in an average leaching rate of 55 kg N ha$^{-1}$ yr$^{-1}$. Despite high nitrogen deposition and low base saturation three plots have almost negligible N leaching (GW, SW, BU). Interestingly enough these plots stand out with a very high crown transparency (proportion of Norway spruce with >25% transparency in 2016, 2017 and 2018 was: 81%, 74% and 43%, respectively, while the average proportion in all 76 Norway spruce plots was 23.4%).

In the mixed regression model, significant predictors for N leaching rates were N deposition, drought, tree removal and water holding capacity (Table 5, Fig 7). Drought predictors included potential evapotranspiration, site water balance and the amount of leached water. The effect of tree removal was largest one year after the removal (lag 1, Table 5). The combined effect of annual tree removal rates (Fig 7B) was calculated as weighted average based on the model coefficients of lag 0, 1, 2 and 3 (Table 5; S4 Table in S1 File). The effect of tree removal is particularly well represented one site (LUB) with a considerable high leaching rate with N inputs of approximately 17 kg N ha$^{-1}$ yr$^{-1}$ (red points in Fig 7A). The reason for these results is a bark beetle infestation in the years 2015 and 2016. N leaching was reduced in dry years and on soils with a high water holding capacity (S6 Fig in S1 File). The C:N ratio in the forest floor was not a significant predictor for N leaching.

Since both N leaching and N deposition decreased during the observation period a direct causal link between these two parameters seems obvious. This hypothesis was tested by extracting annual predictions for the variations in climate (potential evapotranspiration, minimum site water balance and leaching water), or N deposition based on the mixed regression model (Table 5). These estimates were compared with values predicted from the full model and with observed data. This analysis revealed that changes of N deposition and climatic factors contributed equally to the changing N leaching rates (Fig 8).

Although N leaching in plots with coniferous trees was clearly larger than in neighboring plots with deciduous trees (S5 Fig in S1 File), the effect of the proportion of coniferous trees on N leaching revealed not to be significant in the regression analysis. This could be due to a possible confounding effect, since the modeled N deposition itself depends on the tree species.

## 4 Discussion

The present findings of the long-term Intercantonal Forest Observation Program indicate that soil acidification continues to be an important issue in Swiss forests, both in terms of extent

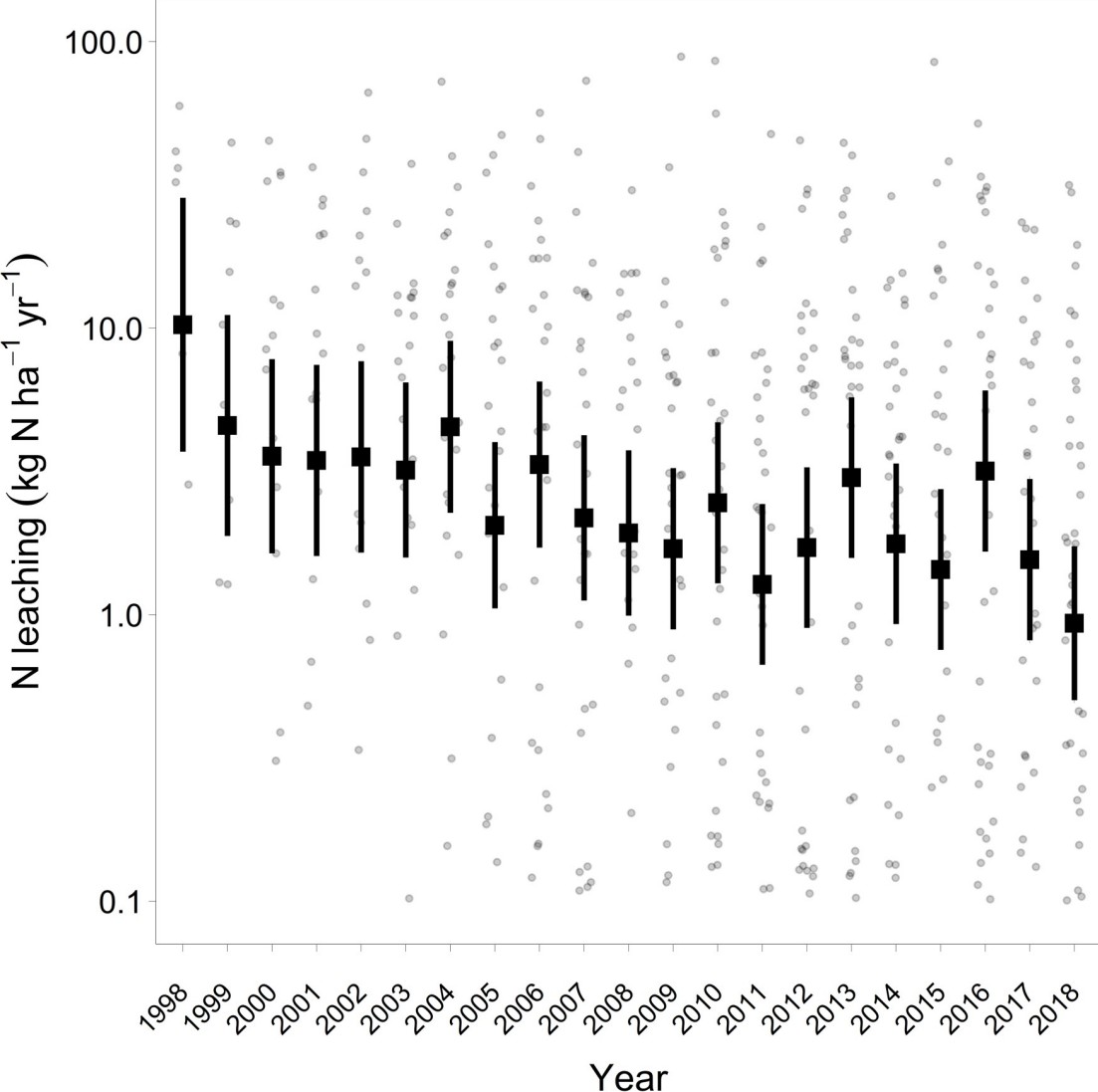

**Fig 6. Development of N leaching 1998–2018.** The decrease with time is significant with p<0.001. Thick squares and error bars (95% confidence intervals) are estimates corrected for the varying number of plots per year from mixed regression (S3 Table in S1 File). Small dots are raw measurements, n = 586.

and ongoing progression. Overall, these results are in accordance with findings reported from ICP Forests in Switzerland [14]. At European level a nonsignificant decrease in the ratio between BC and $Al_{tot}$ was observed, but no signs of recovery from the decreasing acid deposition [6]. The authors explain this by delayed responses. For example, they found that $NO_3^-$ slightly decreased in the subsoil but not in the topsoil.

The thresholds observed for the development of the BC/Al ratio under the current deposition regime (Fig 5) can partially be explained by the soil chemical equilibria of $CaCO_3$ and Al in soils. The pH of 5.9 is rather close to the pH value at which free $CaCO_3$ disappears and the Ca buffer range changes into the cation exchange buffer range (pH<6.2). Between pH 6.2 and 4.2 soils are buffered by the silicate and the cation exchange buffer range which have a lower capacity. At a pH of 4.2 the aluminum buffer range is reached which has a high capacity [66]. The results presented here underline the effectiveness of the various buffer mechanisms. No

**Table 5. Mixed regression model of N leaching with annual data.** Dependent variable: N leaching in kg N ha$^{-1}$ yr$^{-1}$, log transformed. Pseudo-R$^2$ fixed effects = 0.37, Pseudo-R$^2$ including random effect = 0.77, n = 586.

|  | coeff | SE | p-value |
|---|---|---|---|
| (Intercept) | 4.19 | 1.7 | 0.013 |
| N deposition (kg N ha$^{-1}$ yr-1) | 0.10 | 0.02 | <0.001 |
| tree removal rates current year (lag 0) | 0.95 | 0.6 | 0.101 |
| tree removal rates previous year (lag 1) | 2.59 | 0.6 | <0.001 |
| tree removal rates two years before (lag 2) | 1.96 | 0.6 | <0.001 |
| tree removal rates three years before (lag 3) | 1.09 | 0.5 | 0.028 |
| potential evapotranspiration (mm) | -0.003 | 0.001 | <0.001 |
| minimum site water balance (mm) | -2.09 | 0.8 | 0.009 |
| rate of seepage water (mm) | 0.0011 | 0.26 | <0.001 |
| water holding capacity (mm) | -0.01 | 0.004 | 0.001 |

significant relation was found between N deposition and the speed of acidification indicated by the change in BC/Al ratio. This may be partly explained by the various initial soil acid-base-states (different buffer ranges) of the sites in this study. Under homogeneous conditions a clear relation was found between addition of $NH_4NO_3$ and the decrease of BC/Al on a soil with low buffer capacity [43]. The time between changes in deposition and changes in the soil solution depends on the chemical state of the soil and the amount of deposition [7].

Effects of soil acidification on the forest health found in the framework of the long-term Intercantonal Forest Observation Program in Switzerland were reduced rooting depth [20], increased uprooting of trees [67] and increased Mg deficiency [68]. The uprooting of trees was considerably higher on soils with a base saturation <40%. Visible Mg deficiency has strongly

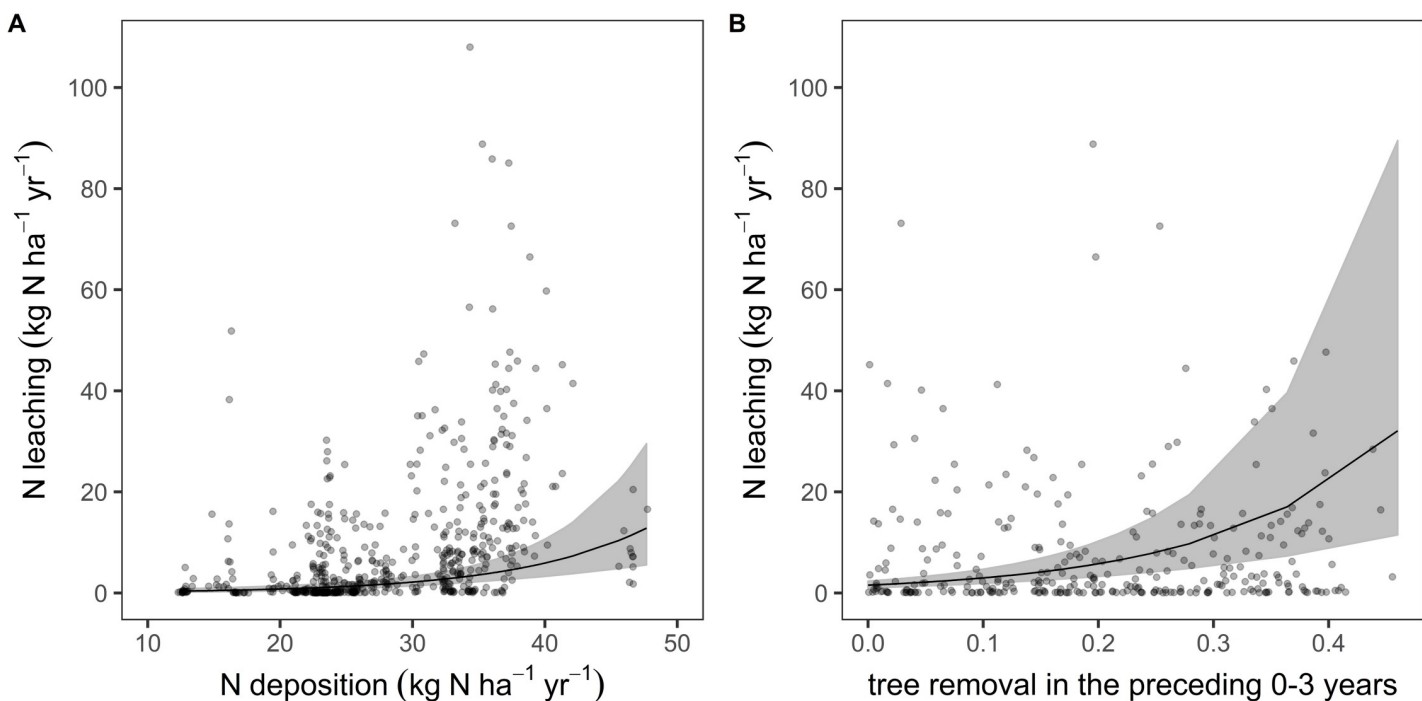

**Fig 7.** A: Relationship between N leaching and N deposition (Table 5). B: Relationship between N leaching and tree removal averaged over 0–3 years (S4 Table in S1 File). Tree removal is in fraction of 1, i.e. 0.4 means that 40% of the trees are removed.

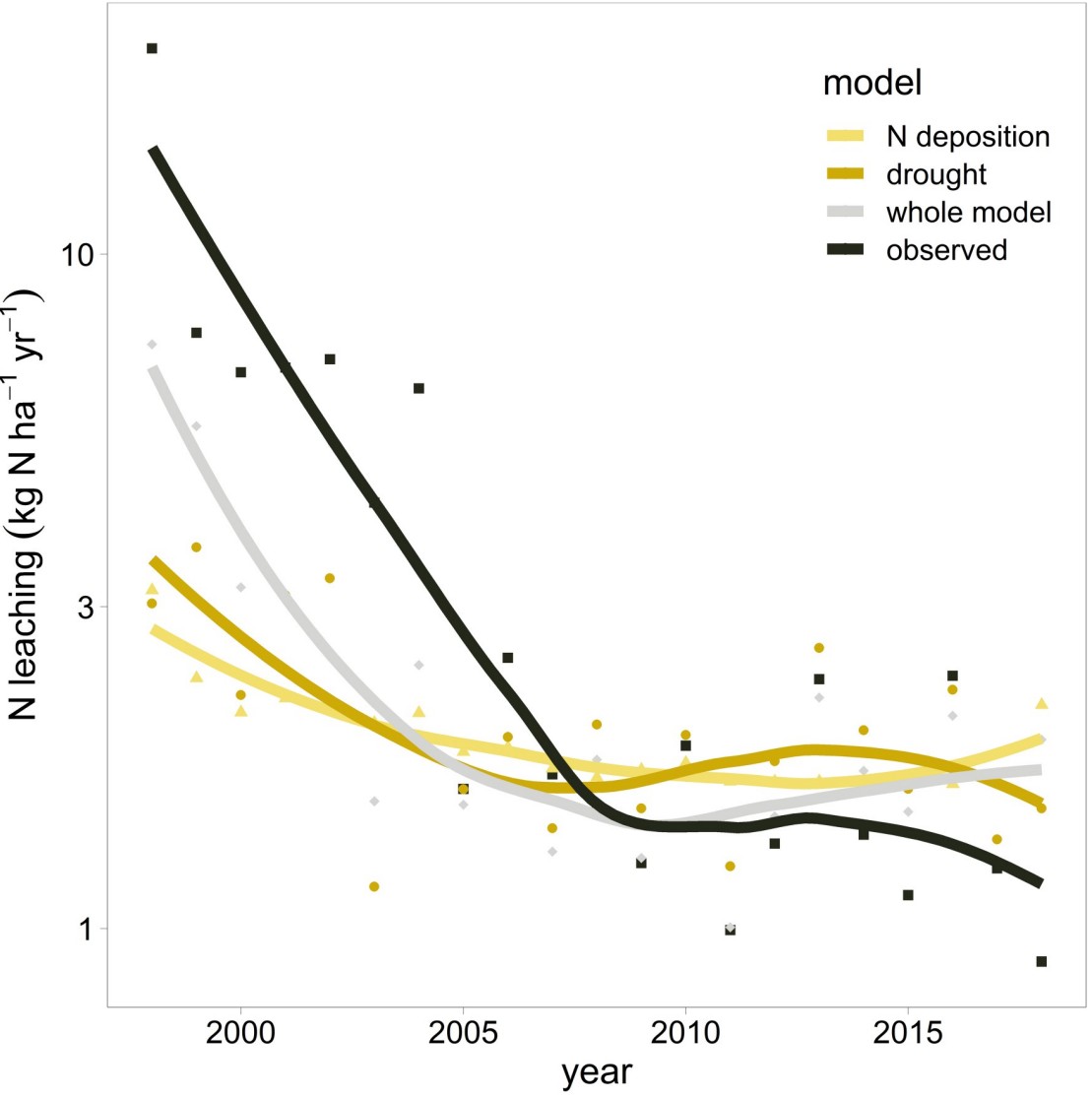

**Fig 8. Comparison of model predictions for the trend of N leaching over time.** The "full model" included all predictors given in Table 4. The "drought" model included predictions for precipitation (leaching water) and temperature (potential evapotranspiration, site water balance). The "N deposition" model predicted values for the changes in N deposition. The model "observed" included the raw data of N leaching. The points are annual means with a loess smoother (degree of smoothing α = 0.8) as lines.

increased in the last 10–15 years, indicating the importance of soil acidification processes for forest health. Moreover, it has been shown that the foliar Mg concentrations in beech leaves are related to the Mg concentrations in the soil solution [68].

The reduction of N deposition between 1998 to 2018 could explain the observed decrease in N leaching. However, the statistical analysis revealed that climate, especially potential evapo-transpiration and runoff, plays an equally important role in this trend (Table 5). Analysis of the temporal variation in N leaching showed a significant contribution of relative mortality or tree removal, that can be detected up to three years after the event. This is in line with previous studies that found an increase in N leaching after clear cutting in a catchment area [37] or after tree removal [69]. In contrary other findings [35], the C:N ratio was not a significant predictor

for N leaching. However, it can be explained by the low C:N ratio of the soils examined. 197 out of 212 plots have a C:N ratio of <25 which was assigned by Gundersen et al. [35] as a threshold for an enhanced leaching risk. Sampling of these soils started when N deposition had been high for a long time.

The higher N leaching under Norway spruce compared to beech, observed on paired plots, is consistent with the observations from Germany [70], [71]. This may be explained by the higher N deposition in Norway spruce stands due to the higher leaf area, the higher surface roughness and the evergreen needles. However, our analyses revealed that tree species was confounded with N deposition as tree species influences deposition modeling.

The present results do not allow general conclusions to be drawn about suitable indicators of N eutrophication. Accepted indicators for eutrophication are increased N leaching, increased nitrate concentration in the soil solution, decreased C:N ratio of the forest floor or increased N in foliage [72], [15]. Based on our results we question the reliability of the N concentration of the leaves as an indicator of eutrophication, since our measurements show that in beech leaves today they are no longer correlated with N deposition, which was the case in the 1980s [38]. The generally low C:N ratios of the soils presented in this study allow no further differentiation. N leaching is elevated on many plots with high N inputs but there are clear outliers: on three plots with a very high N input there is almost no detectable nitrate concentrations in the soil solution and thus no N leaching. None of the commonly accepted eutrophication indicators mentioned above apply to all plots. This conclusion is confirmed by [73]. It must be stated that the confidence interval for the relation between N leaching and N deposition becomes very lager at higher N inputs due to the different site conditions with respect to soils and climate.

## 5 Conclusions

The large number of soil solution measurements of the long-term Intercantonal Forest Observation Program has shown an increase in acidification in most sites between 2005 and 2018, even for base rich soils. The progression of acidification depends on the chemical status of the soil, which is reflected in the buffer ranges. Strongly acidified soils lie in the aluminum buffer range and are therefore less susceptible to further changes of the pH value. The main driver of the observed acidification is high N deposition, which leads to a high nitrate leaching and thus to a high cation loss. In nutrient balance calculations, these cation leaching losses were the most important contribution to the budget, exceeding the input of Ca by weathering and deposition and thus endangering forest sustainability [23].

Soil acidification has negative consequences for forest health, such as increased risk of windthrow on soils with low base saturation <40% [67] or decreased rooting depth for soils with a base saturation <20% [20]. Here we present that, based on the relation between BC/Al ratio and base saturation, these base saturation thresholds translate to a BC/Al ratio in soil solution of 51 and 12, respectively. The BC/Al ratio of 12 is close to the ratio of 10 recommended by [49] as critical limit in critical loads calculations. The relation presented here allows realistic estimates of the relation between BS, pH and BC/Al ratio in mineral soils under field conditions.

The high N deposition above the critical loads is still affecting most of the observed plots, although air pollution measures have resulted in a decrease since the 1980s. The current study provides information to disentangle the effect of drought and nitrogen input on the N leaching losses. It furthermore quantifies the effect of forest management and tree mortality on the variation of N leaching over time. Future research on the interaction between soil solution and forest health should take into account the long-term effects of drought on tree nutrient uptake and changes in ground vegetation.

## Supporting information

**S1 File.**
(PDF)

## Acknowledgments

We would like to thank the field and lab team of the Institute for Applied Plant Biology who performed the monthly sampling and analysis with a lot of patience and care: Delphine Antoni, Dieter Bader, Ute Schröder, Moïse Groelly, Caroline Stritt und Roland Woëffray. Part of the data analysis on N leaching was performed in a project on "Forest and Climate" supported by the Federal Office for the Environment in cooperation with Peter Waldner, WSL.

## Author Contributions

**Conceptualization:** Sabine Augustin.

**Investigation:** Sabine Braun.

**Validation:** Sabine Braun.

**Writing – original draft:** Sabine Braun.

**Writing – review & editing:** Simon Tresch.

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
