## [Decision Letter · Decision Letter 0]

13 Feb 2020

PONE-D-19-35297

Soil solution in Swiss forest stands: a 20 year's time series

PLOS ONE

Dear Dr. Braun,

Thank you for submitting your manuscript to PLOS ONE. After careful consideration, we feel that it has merit but does not fully meet PLOS ONE’s publication criteria as it currently stands. Therefore, we invite you to submit a revised version of the manuscript that addresses the points raised during the review process.

The two reviewers have very different comments on your manuscript. Reviewer 2 is more critical of the manuscript and raise important concerns regarding the objectives, context, title and presentation of the data. It is an impressive data set, but the objectives of the paper should be clear, and the analysis should support the objectives. I have recommended major revisions. Please ensure you address all reviewer comments in the revised manuscript.

We would appreciate receiving your revised manuscript by Mar 29 2020 11:59PM. To enhance the reproducibility of your results, we recommend that if applicable you deposit your laboratory protocols in protocols.io, where a protocol can be assigned its own identifier (DOI) such that it can be cited independently in the future. For instructions see: http://journals.plos.org/plosone/s/submission-guidelines#loc-laboratory-protocols

We look forward to receiving your revised manuscript.

Kind regards,

Julian Aherne

Academic Editor

PLOS ONE

Additional Editor Comments (if provided):

The two reviewers have very different comments on your manuscript. Reviewer 2 is more critical of the manuscript and raise important concerns regarding the objectives, context, title and presentation of the data. It is an impressive data set, but the objectives of the paper should be clear, and the analysis should support the objectives. I have recommended major revisions.

Journal Requirements:

"The author acknowledges the financial support by the Federal Office for the Environment and by the cantons AG, BE, BL, BS, FR, SO, TG, ZH and the environmental offices of the cantons in Central Switzerland."

"SB Federal Office of the Environment Switzerland (measurements and manuscript writing)"

"NO"

Reviewers' comments:

Reviewer's Responses to Questions

**Comments to the Author**

1. Is the manuscript technically sound, and do the data support the conclusions?

Reviewer #1: Yes

Reviewer #2: No

2. Has the statistical analysis been performed appropriately and rigorously? 

Reviewer #1: Yes

Reviewer #2: No

3. Have the authors made all data underlying the findings in their manuscript fully available?

Reviewer #1: No

Reviewer #2: No

4. Is the manuscript presented in an intelligible fashion and written in standard English?

Reviewer #1: Yes

Reviewer #2: No

5. Review Comments to the Author

Reviewer #1: This is a highly relevant and thoroughly elaborated.

It is quite interesting to read that the soil solution monitoring has been continued for 20 years and the future ahead.

The technical description of the approach and the discussion are very well written. The quality of the text is high and it could be published as it is.

Nevertheless, there is always room for improvement:

1/ The introduction of the manuscript is reflecting the state of knowledge of 30 years ago. At that time there was the concern that acidification threatens European forests, and 20 years ago it was stated that N eutrophication is a major concern. There have been many papers since them that put these concerns into perspective. A good source of information of such reviews are the TAMM Reviews of Forest Ecology and Management. - I recommend to either state in the Introduction that the text is reflecting the knowledge when the monitoring programmes were initiated, or (preferred) to expand the Introduction by more recent references.

2/ The methods section is quite complex. The author repeatedly refers to extensive manuals where the single steps are described. Presumingly, very few readers will look up these sources. -- Considering that supplementary material comes with the paper, it would be desirable to show the code that had been used. -- Also a statement on confidence in the invidual steps would be interesting. E.g. how well does SAFE reflect weathering in the field?

3/ The full data sets are not available. - If this is the data policy of the institute, a statement on data ownership could be made.

Reviewer #2: The author has evaluated the chemistry of soil solution at a number of Swiss forest sites with respect to their excedance of critical limits for acidity and nitrate. In addition some models were run to relate the critical limits to site, stand, atmospheric inputs and a number of other factors.

First off, the premise of the paper is not sound. Line 28 states ‘Soil acidification is a serious threat’. This is not the case. Soil acidification is a natural process of soil development in temperate climates where there is a precipitation surplus. The objectives are not clearly stated and the results presented e.g. Ca budgets do not correspond to the stated aim on Line 64-66. Line 66 states that soil solution chemistry will be related to forest health but this is not presented. No evidence is presented to indicate that forests in Switzerland are suffering from the effects of acidification from atmospheric deposition of S and N. Is there forest health data available? Foliar chemistry for example? Evidence of magnesium deficiency or yellowing of needles?

The title of the paper is 'Soil solution in Swiss forest stands: a 20 year's time series' but no time series are presented.

It is not clear what the purpose of the N assessment is and how it relates to acidification and what it contributes to the paper. The author states that ‘N leaching may not always be a good eutrophication indicator’ [L22-23]. This has already been well established in the literature

e.g. Lovett, G.M., Goodale, C.L. A New Conceptual Model of Nitrogen Saturation Based on Experimental Nitrogen Addition to an Oak Forest. Ecosystems 14, 615–631 (2011). https://doi.org/10.1007/s10021-011-9432-z

A lot of the literature cited is quite old e.g. refs 11, 16, 18, 19, 25, 27, 30. A lot of new knowledge has been published in relation to the issue of soil acidification by atmospheric deposition that the author should review.

In addition, a lot of work done on Swiss forest monitoring specifically with respect to this issue, but this work is not cited. I would encourage the author to review these papers and position their work within this context. Sample below

Pannatier, E.G., Thimonier, A., Schmitt, M. et al. A decade of monitoring at Swiss Long-Term Forest Ecosystem Research (LWF) sites: can we observe trends in atmospheric acid deposition and in soil solution acidity?. Environ Monit Assess 174, 3–30 (2011). https://doi.org/10.1007/s10661-010-1754-3

Graf Pannatier, E., Walthert, L. and Blaser, P. (2004), Solution chemistry in acid forest soils: Are the BC : Al ratios as critical as expected in Switzerland?. Z. Pflanzenernähr. Bodenk., 167: 160-168. doi:10.1002/jpln.200321281

Thimonier, A., Graf Pannatier, E., Schmitt, M., Waldner, P., Walthert, L., Schleppi, P., et al. (2010a). Does exceeding the critical loads for nitrogen alter nitrate leaching, the nutrient status of trees and their crown condition at Swiss Long-term Forest Ecosystem Research (LWF) sites? European Journal of Forest Research, 129, 443–461.

Waldner, P., Schaub, M., Graf Pannatier, E., Schmitt, M., Thimonier, A., & Walthert, L. (2007). Atmospheric deposition and ozone levels in Swiss Forests: Are critical values exceeded? Environmental Monitoring and Assessment, 128, 5–17.

I question many aspects of the analysis.

Is Bc:Al expressed as mols of charge here? Is aluminium total Al? What is the basis that an exceedance >1% of the observations [Line 156] is biologically significant?

In calcareous soils, the Al concentrations will be negligible, which makes the Bc:Al ratio very large. This has a spurious effect on the statistical analysis e.g. Figure 2 has Bc:Al values up to 10,000. In the same figure Bc:Al for soil depth >70cm is shown, but Bc:Al is an indicator for stress on fine plant root yet the author states that root depth only extended to 60cm [L173]. So I don’t think Bc:Al below this depth is relevant. For a study of acidification, the analysis should be confined to those soils in the Al buffering pH range.

The regression analysis is not appropriate. It is not clear what the objective of this analysis is and many of the relationships have already been established in the literature. For example, why is Bsat a predictor for Bc:Al in soil solution? Bsat should be considered a response variable and not a predictor in this case. Also Bsat to Bc:Al relationships have already been developed and are used in the critical loads modelling e.g. Gaines-Thomas and Gapon equations.

I would also question the data used as an input to the model. The author states that the number of sites ranges from ‘9 to 47’. I don’t think it is appropriate to have uneven numbers of observations for different sites. Also it is not clear how the 5 year time interval [L186] was generated for predictors. Was soil Bsat, Soil C:N measured every 5 years? Was the weathering rate recalculated for every 5 year interval? Time intervals for a particular site are not independent observations – was this accounted for in the model?

The relationship presented in Figure 2, between pHCaCl2 and Bc:Al is spurious.

I don’t think the results presented in Table 3 are valid. The levels of significance are likely a result of the large number of observations used.

If N leaching has decreased and N deposition is the main driver of acidity [L18-19] then why is the rate of soil acidification increasing [L17-18, L264]? Why wasn’t N deposition significant in the model of soil acidification (Table 3)?

In Fig 3 the ca leaching rate is greater than deposition (and weathering + deposition. What is the proposed driver of this Ca loss, if N deposition is decreasing? How do these values relate to other published values for weathering? Was the weathering model calibrated? It has previously been reported that soil solution in acid pseudogleyed horizons can be influenced by underlying clay soils (Graf Pannatier 2004). Was this taken into account in the analysis?

In relation to Fig. 5 it is not clear what this figure is trying to show. Why not simply plot Bc:Al change with time? Again the linear relationships don’t seem valid.

The statistical relationships in figures 6 to 8 don’t seem valid, but seem to be driven by a few outliers. Why does N leaching extend above 150KgN?

The English needs to be improved and there are numerous grammatical errors and formatting errors. This makes the paper difficult to read. some examples..

L374 comma at the end of line

L13; ‘ranging’ instead of ‘growing’

L18: ‘Main driver’ – should be ‘The main driver..’

L36 'concerns on forest health'..should be 'concerns about forest health'

The conclusions are not supported by the findings.

6. PLOS authors have the option to publish the peer review history of their article (what does this mean?). If published, this will include your full peer review and any attached files.

Reviewer #1: Yes: Robert Jandl

Reviewer #2: No

---

## [Author Response · Author response to Decision Letter 0]

30 Mar 2020

Reviewer #1:

5. Review Comments to the Author

Reviewer #1: This is a highly relevant and thoroughly elaborated.

It is quite interesting to read that the soil solution monitoring has been continued for 20 years and the future ahead.

Reviewer #1: The technical description of the approach and the discussion are very well written. The quality of the text is high and it could be published as it is.

Reviewer #1: Nevertheless, there is always room for improvement:

1/ The introduction of the manuscript is reflecting the state of knowledge of 30 years ago. At that time there was the concern that acidification threatens European forests, and 20 years ago it was stated that N eutrophication is a major concern. There have been many papers since them that put these concerns into perspective. A good source of information of such reviews are the TAMM Reviews of Forest Ecology and Management. - I recommend to either state in the Introduction that the text is reflecting the knowledge when the monitoring programmes were initiated, or (preferred) to expand the Introduction by more recent references.

Author: Thank you for your valuable comments. We changed now the wording in the first sentence to show that soil acidification has been an issue in the 1980's. 

Lines 29-32: Since the 1980s it has been recognized that soil acidification due to anthropogenic input of sulfur and nitrogen compounds ([1]) poses a serious threat to forest health ([2]). An increase in soil acidification, related to atmospheric acid deposition, has been reported in many European countries such as Germany ([3]), Sweden ([4] or France ([5]).

However, the later sentences have already stated that mitigation measures have improved the situation. Therefore, it is necessary to refer to older literature as they are the basis of the Critical Loads for Acidity. I do, however, not agree with the conclusions of the proposed Tamm review by Högberg [6] that there is no N problem as they refer to Northern Ecosystems where total N deposition is much lower and the economy of forest growth and thus biomass production are much more important. In Switzerland, forest N eutrophication is a problem, not a fertilization measure.

Reviewer #1: 2/ The methods section is quite complex. The author repeatedly refers to extensive manuals where the single steps are described. Presumingly, very few readers will look up these sources. -- Considering that supplementary material comes with the paper, it would be desirable to show the code that had been used. -- Also a statement on confidence in the invidual steps would be interesting. E.g. how well does SAFE reflect weathering in the field?

Author: We have now added a description on base saturation determination as this is the part which the reader should know without looking up – is this conclusion correct? We have added the R scripts we used to the supplementary material and a remark on validation of SAFE calculations to the methods section.

Reviewer #1: 3/ The full data sets are not available. - If this is the data policy of the institute, a statement on data ownership could be made.

Author: We have uploaded all R codes and a description of data manipulation and models including the raw data on the data base dryad:

Data Availability: The data and an R project including an RMarkdown file with all R codes, model outputs and diagnostic plots have been uploaded to Dryad: https://doi.org/10.5061/dryad.2z34tmphm.

Before acceptance of the paper the data are accessible via:

https://datadryad.org/stash/share/PjiHqj7uwJK13MBFbbbiRVeoO_DI1byCEPpHDD3bDms

 

Reviewer #2:

Reviewer #2: The author has evaluated the chemistry of soil solution at a number of Swiss forest sites with respect to their excedance of critical limits for acidity and nitrate. In addition some models were run to relate the critical limits to site, stand, atmospheric inputs and a number of other factors.

Reviewer #2: First off, the premise of the paper is not sound. Line 28 states ‘Soil acidification is a serious threat’. This is not the case. Soil acidification is a natural process of soil development in temperate climates where there is a precipitation surplus. The objectives are not clearly stated and the results presented e.g. Ca budgets do not correspond to the stated aim on Line 64-66. Line 66 states that soil solution chemistry will be related to forest health but this is not presented. No evidence is presented to indicate that forests in Switzerland are suffering from the effects of acidification from atmospheric deposition of S and N. Is there forest health data available? Foliar chemistry for example? Evidence of magnesium deficiency or yellowing of needles?

Author: Thank you for your comments. According to your suggestions we changed the first sentences of the introduction:

Lines 29-34: Since the 1980s it has been recognized that soil acidification due to anthropogenic input of sulfur and nitrogen compounds ([1]) poses a serious threat to forest health ([2]). An increase in soil acidification, related to atmospheric acid deposition, has been reported in many European countries such as Germany ([3]), Sweden ([4] or France ([5]). Due to mitigation measures, the deposition of acidifying substances in Europe, in particular of sulfur compounds, has decreased in recent years, but eutrophication due to the deposition of reactive nitrogen compounds remains a critical issue ([7], [8]). 

The reader can now understand that measures have been taken against acid deposition. It is correct that soil acidification is a natural process but the pH reached with the main natural acidifying substance HCO3- is +/- pH 4.8-5.0, depending on the partial pressure of the CO2. In the uppermost cm of the mineral soil the organic acids can acidify the soil somewhat more, but the organic acids cannot reach the subsoil, because they dissociate in the Bs-horizon of a soil, at the latest (due to the normal pH gradient in soils). In contrast, the acidification due to the input of the mobile anions SO42- and NO3- is more “effective”; SO42- can be stored in the B horizon, but this is not stable, the SO42-- desorbs if the S-input declines. In this case, the deeper soil horizons acidify. This was demonstrated in a lot of intensive monitoring plots. Nitrate cannot be stored in the soil. If it is not taken up by vegetation, the NO3- passes through the profile, together with a concomitant cation (BC or an acid cation like Mn or Al), diminishing the BS% in soils. We have added now a reference to [1] to highlight these processes.

The relations between soil acidification and forest health have been discussed in the submitted manuscript on line 50 ([9]) and line 416 ([10] ). These two references show results from the forest plots presented in this study but we agree that this has not been stated clearly. We have now added a separate paragraph to the Discussion section including information on Mg deficiency which has also increased in recent years, and that foliar Mg has been related to soil solution Mg.

Lines 410-416: Effects of soil acidification on forest health in the forest plots presented in the current study have been observed on rooting depth [9], uprooting of trees [10] and on Mg deficiency [11]. The uprooting of trees was clearly increased on soils with a base saturation <40%. Visible Mg deficiency also has strongly increased within the last 10-15 years, suggesting that soil acidification is still an issue in the Intercantonal forest observation plots. Foliar Mg concentrations in beech leaves have been shown to be related with Mg concentrations in soil solution [11].

Reviewer #2: The title of the paper is 'Soil solution in Swiss forest stands: a 20 year's time series' but no time series are presented.

Author: This is not true. Figure 4 shows a time series of the BC/Al ratio and Figure 6 on N leaching. But with 47 plots it is not feasible to show all plots separately. The graphs for all single plots are, however, accessible through the project report ([12] ).

Reviewer #2: It is not clear what the purpose of the N assessment is and how it relates to acidification and what it contributes to the paper. The author states that ‘N leaching may not always be a good eutrophication indicator’ [L22-23]. This has already been well established in the literature

e.g. Lovett, G.M., Goodale, C.L. A New Conceptual Model of Nitrogen Saturation Based on Experimental Nitrogen Addition to an Oak Forest. Ecosystems 14, 615–631 (2011). https://doi.org/10.1007/s10021-011-9432-z

Author: The reason why N is related with acidification is that the negatively charged nitrate carries along positively charged ions. We have added a remark to the corresponding paragraph. We have also introduced a reference to Lovett & Goodale but their conclusion is not that N leaching is not a eutrophication indicator but that it may occur simultaneously with other stages of N saturation. The sentence that "N leaching may not always be a good eutrophication indicator" is in the abstract and cannot refer to a reference at this place.

Reviewer #2: A lot of the literature cited is quite old e.g. refs 11, 16, 18, 19, 25, 27, 30. A lot of new knowledge has been published in relation to the issue of soil acidification by atmospheric deposition that the author should review.

Author: We introduced a number of newer references ([8], [13], [14],[15], [16] ). However, when referring to the Critical Load manuals it is necessary to cite the old literature which was the basis to setting the limits.

Reviewer #2: In addition, a lot of work done on Swiss forest monitoring specifically with respect to this issue, but this work is not cited. I would encourage the author to review these papers and position their work within this context. Sample below

Pannatier, E.G., Thimonier, A., Schmitt, M. et al. A decade of monitoring at Swiss Long-Term Forest Ecosystem Research (LWF) sites: can we observe trends in atmospheric acid deposition and in soil solution acidity?. Environ Monit Assess 174, 3–30 (2011). https://doi.org/10.1007/s10661-010-1754-3

Graf Pannatier, E., Walthert, L. and Blaser, P. (2004), Solution chemistry in acid forest soils: Are the BC : Al ratios as critical as expected in Switzerland?. Z. Pflanzenernähr. Bodenk., 167: 160-168. doi:10.1002/jpln.200321281

Thimonier, A., Graf Pannatier, E., Schmitt, M., Waldner, P., Walthert, L., Schleppi, P., et al. (2010a). Does exceeding the critical loads for nitrogen alter nitrate leaching, the nutrient status of trees and their crown condition at Swiss Long-term Forest Ecosystem Research (LWF) sites? European Journal of Forest Research, 129, 443–461.

Waldner, P., Schaub, M., Graf Pannatier, E., Schmitt, M., Thimonier, A., & Walthert, L. (2007). Atmospheric deposition and ozone levels in Swiss Forests: Are critical values exceeded? Environmental Monitoring and Assessment, 128, 5–17.

Author: We have added the Pannatier 2011 reference..

Reviewer #2: I question many aspects of the analysis.

Is Bc:Al expressed as mols of charge here? Is aluminium total Al? What is the basis that an exceedance >1% of the observations [Line 156] is biologically significant?

In calcareous soils, the Al concentrations will be negligible, which makes the Bc:Al ratio very large. This has a spurious effect on the statistical analysis e.g. Figure 2 has Bc:Al values up to 10,000. In the same figure Bc:Al for soil depth >70cm is shown, but Bc:Al is an indicator for stress on fine plant root yet the author states that root depth only extended to 60cm [L173]. So I don’t think Bc:Al below this depth is relevant. For a study of acidification, the analysis should be confined to those soils in the Al buffering pH range.

Author: The calculation of BC/Al is described in the UNECE manuals but we have now inserted an explanation how we did it. The 1% limit was used to remove outliers. The mapping manual does not define how many values have to exceed the threshold but the roots will react to single values rather than any average. We agree that calculation of BC/Al in calcareous soils is subjected to errors but omitting these values would bias the dataset towards acid soils. Looking at outliers in the graphs is misleading as we have log transformed them before data analysis. We agree that BC/Al at soil depth >70 cm is not relevant for plant roots but we am showing these values for the sake of completeness, to show the trends in soil solution chemistry. We do not draw any plant relevant conclusions on these values.

To confine acidification studies only on soils with pH 3.8 – 4.2 (Al buffer range) would lead to wrong conclusions; to detect trends one should use the whole existing pH-range. Thus, we do not agree with the recommendation that the data analysis should be restricted to the Al buffer range.

Reviewer #2: The regression analysis is not appropriate. It is not clear what the objective of this analysis is and many of the relationships have already been established in the literature. For example, why is Bsat a predictor for Bc:Al in soil solution? Bsat should be considered a response variable and not a predictor in this case. Also Bsat to Bc:Al relationships have already been developed and are used in the critical loads modelling e.g. Gaines-Thomas and Gapon equations.

Author: The aim of the regression analysis was to find out why the decrease in BC/Al is going on fast in some plots and slow in others. We have added now an introducing sentence. The Gaines-Thomas and Gapon equations are useful in soil chemical models to describe the exchange of cations between soil solution and soil solid phase. Here we follow a statistical approach to explain the measured changes in the composition of the soil solution. For comparisons of the critical thresholds of BC/Al in soil solution with the base saturation of the soil solid phase it is important to have field data for validation

Reviewer #2: I would also question the data used as an input to the model. The author states that the number of sites ranges from ‘9 to 47’. I don’t think it is appropriate to have uneven numbers of observations for different sites. Also it is not clear how the 5 year time interval [L186] was generated for predictors. Was soil Bsat, Soil C:N measured every 5 years? Was the weathering rate recalculated for every 5 year interval? Time intervals for a particular site are not independent observations – was this accounted for in the model?

Author: The mixed regression accounts for unequal number of observation and for unequal replication. So from the statistical point of view it is not a problem to compare Bsat and C:N measured once or weathering rate modelled once with monthly measured soil solution data. The mixed regression handles also repeated measures from one site, e.g. time intervals.

The Bsat and soil C:N were not measured every 5 year, because it is not expectable to find measurable changes in such a short time, esp. in a heterogeneous environment like forest soils. The weathering rate is stable, it is enough to measure it once. In contrast, the composition of the soil solution is a fast reacting monitor of changes, that’s why it is measures monthly, and for the annual changes aggregated to annual means.

Reviewer #2: The relationship presented in Figure 2, between pHCaCl2 and Bc:Al is spurious.

Author: Figure 2 shows measured BC/Al ratio in relation to pH and base saturation. Based on this comment it is not clear how this is wrong or how it could be changed or improved.

Reviewer #2: I don’t think the results presented in Table 3 are valid. The levels of significance are likely a result of the large number of observations used.

Author: Figure 5 illustrates the regressions with the confidence intervals and the data points so the reader can check that the regressions are quite good and not just an artefact of the high number of points.

Reviewer #2: If N leaching has decreased and N deposition is the main driver of acidity [L18-19] then why is the rate of soil acidification increasing [L17-18, L264]? Why wasn’t N deposition significant in the model of soil acidification (Table 3)?

Author: The processes leading to soil acidification are not linear so a decrease in N deposition (which is actually small anyway in Switzerland) will not necessarily lead to an immediate decrease in soil acidification (for recovery effects see e.g. Verstraeten 2017). N is still accumulating as long as the deposition of N is larger than the use by the forest stand. The relation between the amount of N deposition and soil acidification can be shown in the experiment but in the field the geological heterogeneity is very large. We have now made a reference to this experiment.

Reviewer #2: In Fig 3 the ca leaching rate is greater than deposition (and weathering + deposition. What is the proposed driver of this Ca loss, if N deposition is decreasing? How do these values relate to other published values for weathering? Was the weathering model calibrated? It has previously been reported that soil solution in acid pseudogleyed horizons can be influenced by underlying clay soils (Graf Pannatier 2004). Was this taken into account in the analysis?

Author: The deposition in the X-axis is Ca-deposition, not N deposition. Thus, Figure 3 does not show N deposition but the Ca leaching is the result of NO3 -leaching. 

The weathering model was calibrated in Sweden. Recent analyses with Al in tree rings confirm that the dynamic soil acidification model is performing quite well. We have put this remark to the methods section. – The indicated reference discusses the effect of underlying clay soils on soil solution in 80 cm depth. As the soil solution chemistry at larger depths is not in the focus of the current paper, such processes do not change the main conclusions. Similar situations (soil wetness >2 and calcareous layer present in <100 cm with acid topsoil) occur in just one among 47 plots.

Reviewer #2: In relation to Fig. 5 it is not clear what this figure is trying to show. Why not simply plot Bc:Al change with time? Again the linear relationships don’t seem valid.

Author: A plot of BC/Al change with time is given in Figure 4. Figure 5 illustrates the speed of the change in BC/Al in relation to the predictors in Table 3. Predicted values including 95% confidence intervals are conditioned on all other fixed effects. Negative changes signify an expected decrease in BC/Al. The linearity of the relations was tested using polynomial functions

Reviewer #2: The statistical relationships in figures 6 to 8 don’t seem valid, but seem to be driven by a few outliers. Why does N leaching extend above 150KgN?

Author: The regressions were made with log transformed values (this is stated in the text) in order to meet the assumptions for a liner mixed effect model. The models were tested for robustness. The shaded areas give the 95% confidence interval. For the graphs, the values were backtransformed so the scatter seems larger than it actually was. 

The mixed regression model given in Table 4 explains about 77 % of the variation of N leaching. The amount of n=586 seems valid to test the given amount of fixed effects. These fixed effects explained alone 36% of the data. The Pseudo-R2 are now given in the tables.

E.g. Lines 370-372:

Table 4: Mixed regression model of N leaching with annual data (n=586). Dependent variable: N leaching in kg N ha 1 yr 1, log transformed. Pseudo-R2 fixed effects = 0.37, Pseudo-R2 including random effect = 0.77.

All models can now be seen in the Supplementary Materials in the RMarkdown documented R codes in order to achieve the highest standards of statistical reproductivity of the models.

The points with very high N leaching have been plot with disturbances. The points show annual leaching rates. 

Reviewer #2: The English needs to be improved and there are numerous grammatical errors and formatting errors. This makes the paper difficult to read. some examples.

Author: A great effort has been invested to improve wordings and eliminate all grammatical errors.

L374 comma at the end of line

Author: Thank you for this note this has been changed.

L13; ‘ranging’ instead of ‘growing’

Author: Thank you for this comment: 'ranging' means that they extend from .. to. 'Growing' means that the number has increased. We have now replaced "growing" by "increasing".

L18: ‘Main driver’ – should be ‘The main driver..’

Author: Thank you for this note this has been changed.

L36 'concerns on forest health'..should be 'concerns about forest health'

Author: Thank you for this note this has been changed.

The conclusions are not supported by the findings.

 Author: We cannot comment this as this is a statement without proof.

---

## [Decision Letter · Decision Letter 1]

28 Apr 2020

PONE-D-19-35297R1

Soil solution in Swiss forest stands: a 20 year's time series

PLOS ONE

Dear Dr. Braun,

Thank you for submitting your manuscript to PLOS ONE. After careful consideration, we feel that it has merit but does not fully meet PLOS ONE’s publication criteria as it currently stands. Therefore, we invite you to submit a revised version of the manuscript that addresses the points raised during the review process.

All reviewers agree that this manuscript is an important contribution to the literature, especially given the long-term data set. Further, it is noted that the revised manuscript is much improved. Nonetheless, two of the reviewers have indicated that further revisions are required, specifically with respect to clarification of the research question and hypothesis, presentation of data, and the description of the data analysis. 

We would appreciate receiving your revised manuscript by Jun 12 2020 11:59PM. To enhance the reproducibility of your results, we recommend that if applicable you deposit your laboratory protocols in protocols.io, where a protocol can be assigned its own identifier (DOI) such that it can be cited independently in the future. For instructions see: http://journals.plos.org/plosone/s/submission-guidelines#loc-laboratory-protocols

We look forward to receiving your revised manuscript.

Kind regards,

Julian Aherne

Academic Editor

PLOS ONE

Additional Editor Comments (if provided):

Dear Sabine Braun, please note that your manuscript was sent out for further review as one of the initial reviewers (the more critical of the two) was unavailable. All reviewers agree that this manuscript is an important contribution to the literature, especially given the long-term data set. Further, it is noted that the revised manuscript is much improved. Nonetheless, two of the reviewers have indicated that further revisions are required, specifically with respect to clarification of the research question and hypothesis, presentation of data, and the description of the data analysis. while I have indicated 'Major Revisions' are required in line with the recommendations of the reviewers, I believe that these are minor revisions and that they can be easily addressed. I look forward to seeing your revised manuscript.

Reviewers' comments:

Reviewer's Responses to Questions

**Comments to the Author**

1. If the authors have adequately addressed your comments raised in a previous round of review and you feel that this manuscript is now acceptable for publication, you may indicate that here to bypass the “Comments to the Author” section, enter your conflict of interest statement in the “Confidential to Editor” section, and submit your "Accept" recommendation.

Reviewer #1: All comments have been addressed

Reviewer #3: (No Response)

Reviewer #4: (No Response)

2. Is the manuscript technically sound, and do the data support the conclusions?

Reviewer #1: Yes

Reviewer #3: Yes

Reviewer #4: No

3. Has the statistical analysis been performed appropriately and rigorously? 

Reviewer #1: Yes

Reviewer #3: I Don't Know

Reviewer #4: Yes

4. Have the authors made all data underlying the findings in their manuscript fully available?

Reviewer #1: Yes

Reviewer #3: Yes

Reviewer #4: Yes

5. Is the manuscript presented in an intelligible fashion and written in standard English?

Reviewer #1: Yes

Reviewer #3: Yes

Reviewer #4: Yes

6. Review Comments to the Author

Reviewer #1: I want to congratulate the authors to an exceptional manuscript. The text has a high potential to be used in classrooms. The coverage of the topic is comprehensive. A particular value is the availability of the code of the data evaluation. The description of the data has the same high quality as the main text itself. I am greatly impressed by the quality of the submission and hope that it will find wide recognition.

Reviewer #3: This is an interesting analysis of a 20 year's time series of measured element concentrations in forests soil solution in relation to different environmental variables. The work adds to current knowledge on progressive acidification of forest soils intensified by atmospheric nitrogen and sulphur deposition. The authors elaborated further on a first version of their manuscript taking into account various comments of two reviewers. I think this has substantially improved the overall quality of the paper, but there are still some issues that need to be solved before it can be published, particularly concerning the modeling analysis.

detailed comments

L18: ‘Acidification indicators remained stable at high levels...’ : I think this terminology is too vague for an abstract. Could you please formulate it in more concrete terms?

L23 (and further): The term ‘N leaching’ is used throughout the paper, which I think is not entirely correct, since dissolved organic forms of N (DON) are not included. I suggest to consistently write ‘nitrate leaching’ (assuming that ammonium leaching is negligible).

L25-26: I suggest to drop the following sentence: ‘Therefore, we suggest a restricted use of N leaching as an eutrophication indicator’. Plenty of studies have shown that NO3 leaching is an important indicator of eutrophication/N saturation, but since forests are natural ecosystems with a huge variety in environmental conditions there logically also must be exceptions.

L58-59: as the BC/Al ratio is defined here, I think you should also mention what you mean with base cations (Ca2+ + K+ + Mg2+) in this sentence.

L69-71: I suggest to refer to the groundwater directive here. DIRECTIVE 2006/118/EC OF THE EUROPEAN PARLIAMENT AND OF THE COUNCIL of 12 December 2006 on the protection of groundwater against pollution and deterioration (https://eur-lex.europa.eu/LexUriServ/LexUriServ.do?uri=OJ:L:2006:372:0019:0031:EN:PDF).

L137 and L178: it is not clear to me whether and how the alkalinity data were actually used in your analysis. I assume ANC was calculated rather as the balance between anions and cations? If alkalinity itself was not used, the corresponding part of the sentence in L137 should be removed.

L224 and Fig. 2: a quadratic term was used in order to allow modeling of a non-linear relationship. This approach has not been described in the materials and methods section. I actually wonder if a linear mixed additive model (gamm) applied on the untransformed data would not be more appropriate here, as it would probably be more easy to understand. The current graphs are somewhat misleading: it looks as if the curve flattens of at higher BS, but instead it is the opposite: if the Y-axis would be in regular units instead of log-scale, one would see an exponentially increasing curve.

L226: I do not understand this. BC/Al ratio at the start of the 5 year period was included as a predictor for the BC/Al ratio?

Section 3.1.3 and 3.1.4: I have some doubts on the modeling analysis. The number of independent variables in the initial models (about 10) is actually quite large for the number of sites (n = 47). While there is no general rule of restriction, a large number of variables could easily lead to overfitting of the model and thus wrong interpretation and results. I am not entirely convinced that the applied backward selection with AIC/BIC excluded the least important variables. Instead, it might have been a better approach to restrict the initial number of variables (pre-selection) based on present knowledge or to do a forward selection.

Table 2 and 3: The data for the dependent variables BC/Al ratio and N leaching were log-transformed, as mentioned in the caption of both tables. Log-transformation changes the relationship between the dependent and independent variables and may therefore lead to a wrong interpretation of the results. This choice should thus be clearly motivated, which information is currently missing in the materials and methods section.

L253-254: you wrote that normality and homogeneity of variance of the residuals was checked, but it is necessary to show also graphs of normalised residuals vs. fitted values and of normalised residuals against each independent variable in the model, in order to allow the reader to judge the validity of models. In L258-259 it is stated that this information is included in the Supplementary information, but I couldn’t find it there.

Reviewer #4: This manuscript assesses recent soil acidification in Swiss forest stands in response to atmospheric acidic deposition using soil solution chemistry measured over the last 20 years. This is a valuable dataset, due to the large number of plots that have been monitored for many years in different soil types across Switzerland in two major forest types (spruce and beech) and exposed to different levels of atmospheric deposition.

I was not reviewer of the first version of the manuscript. It seems that the authors have made a great deal of effort in improving the quality of the manuscript. However, I have major concerns about the quality of the revised manuscript: 1) there is no research questions and hypothesis, 2) the description of the current state of knowledge is thin and the authors’ results are not integrated in a broader context, 3) the way of presenting some results is questionable. I explain the three points:

1) Missing research question: the authors do not ask any research question and do not postulate any hypotheses. It is therefore difficult to assess whether their statistical analyses allow them to answer their questions! In recent decades, air pollution reduction policies have resulted in a large reduction in sulphur emissions and, to a lesser extent, in nitrogen emissions. However, the authors do not question the effect of these measures on soil acidification in their introduction. It is not clear whether the focus is on trends of soil acidity indicators or on the effects of forest health.

2) Insufficient description of the current state of knowledge: There are few references to recent studies related to the trends of the soil solution chemistry in response to declining acidic deposition. Yet many publications have been published in the last decade about the potential recovery of soil solution in forest soils and of surface waters in Europe and North America (in particular in countries participating in ICP-Forests and ICP-Waters) in response to declining acidic deposition. In addition, a major European study has been recently published (Johnsson et al. 2018, Global Change Biology, DOI: 10.1111/gcb.14156) on the response of soil solution chemistry in European forests. There was no mention of this study in the revised manuscript. Since the current state of the knowledge is insufficiently described, the introduction does not tell which knowledge gap this manuscript aims to fill. Also in the discussion, the results are not integrated in a larger context, not even at the national scale. There are few references to Swiss and European studies related to soil (solution) acidification.

3) Several concerns about the way of presenting results:

3a. I could not see a systematic way in presenting the data. The level of data aggregation for the different graphs and tables seems to be different, which makes the manuscript heterogeneous and difficult to understand (e.g. analysis of different depth intervals in figures). The tables are also difficult to read because units are not included. For instance, it would help the reader to know which year corresponds to the intercept and the units of intercept and changes (relative, absolute slope?) in the tables presenting the linear models.

3b. The topic of the manuscript is soil solution chemistry over the last 20 years, but I have not learnt much about the quality of the soil solution and how it has changed over time in the studied plots. A statistical summary of concentrations of individual cations and anions and the corresponding trends would have been useful. To present only the BC/Al ratio is insufficient to assess changes in soil solution chemistry. Knowing the behavior of sulphur, base cations and aluminum concentrations (or fluxes if available) is essential to better understand the effect of declining atmospheric deposition. The authors refer to a report illustrating time series at each plot. The link in the reference list is, however, not valid. After searching in Internet, I found this interesting report. The graphs show that the temporal trends differ between the plots and depths and, interestingly, that acidity indicators are stabilizing in the last years in some plots after a decrease in the first years of observation.

3c. Table 2: this analysis is not convincing. The authors mean in this graph that even a low percentile of measurements below the critical thresholds at a given plot would have a critical impact, which has not been reported in the literature. A more detailed analysis including information about the distribution of BC/Al values would be more informative.

3d. Figure 2: The calculation of BC/Al ratio at pH > 6.5 is misleading. The aluminum concentrations are very low, close to the detection limit and therefore cannot be quantified precisely. Very small differences in Al concentrations lead to large variations in BC/Al. In that respect, the quantification limit of Al should be reported to assess the uncertainty related to high BC/Al. In addition, inorganic Al was measured as difference before and after passing the samples through an ion exchanger and therefore uncertainties add up. Another tricky point in this figure is the comparison of soil and soil solution parameters using large depth intervals (<70 cm and >70 cm). The authors do not explain how BC/Al, BS and pH measurements were aggregated in this large depth interval. In this analysis, the presence of calcareous parent material in the subsoil might explain the discrepancies between BS (Fig. 2B) and soil pH (Fig. 2C) (see Blaser et al. 2008, https://doi.org/10.1002/jpln.200625213). Acidified forest soils on calcareous parent material in Switzerland usually have a strong base saturation gradient with depth. Also soil water regime (hydromorphy) and reduced drainage play an important role in the chemistry of these soils.

3e. Figure 3 is very interesting, because few studies have assessed Ca weathering rates and compare them to Ca leaching. However, information about temporal trends is missing. It is known from many studies that concentrations of base cations in soil solution have decreased in the last decade. The driver responsible for this decrease is not completely clear (chemical equilibrium due to declining sulphate concentrations in soil solution? declining atmospheric deposition of base cations? increase in root uptake? ). The temporal analysis of BC leaching from the soil is therefore also relevant because the rate of BC loss from the soil might have slowed down in the last decade, which is still a relevant information to assess the effect of air pollution reduction measures.

3f. Figure 5 is interesting. Fig. 5A shows that high BC/Al ratios at a given time are likely to decrease in the following years, while low BC/Al ratios are likely to stabilize or even increase. In lign 312, the authors write “The decreasing trends were getting weaker the stronger the soils are acidified”. I do not understand why the predicted values in 5B and 5C actually show the opposite trend.

Other specific comments:

• l. 17: remove “in Swiss forest soils”, not necessary

• l.18. “remained stable at high level”: formulation is not clear

• l. 20 “an increasing acidification”: not clear. An increasing acidification rate? Acidification is happening anyway.

• l. 28-29: “Taken together, this study provides evidence of increasing soil acidification in Swiss forest stands.” This is not clear. Do the author mean that the rates of acidification are increasing? Fig. 5A illustrates a more balanced view.

• l.31: “climate change”: meteorological parameters were used in the analysis. Climate change was not analysed.

• l.45-48: The link between the exceedance of critical loads for acidity and the decreasing BC/Al ratio in Pannatier et al. (2011) is not clear. Declining BC/Al does not mean that critical loads are exceeded.

• L.51: “is of crucial importance”: why?

• L. 132-133: suction cups? at which depth? Basic information on the type of suction cups used in this study and sampled depths should be available in this section.

• L. 159: what are the vegetation parameters?

• L. 211: reference in prep is not suitable

• L. 289: Reference to buffer ranges is missing

• L. 318, l. 470: not comma before that. Remove everywhere

• L. 401: “The results are confirmed by findings of the long-term forest monitoring of ICP Forests in Switzerland ([66]).” This sentence is not correct since ref 66 is much older.

• L. 445-447: “Based on our results we question the reliability of the N concentration of the leaves as an indicator for eutrophication, since our measurements show that in beech leaves today they are no longer correlated with N deposition, which was the case in the 1980s”. Not shown by data from this manucript.

• L. 476-L. 475. “Soil acidification has negative consequences for forest health (…). This conclusion is not supported by the findings of this manuscript.

• L. 479: no analysis of climate change in this manuscript. Meteorological parameters were used. The manuscript did not show that the droughts were related to climate change. References are needed to make this link.

7. PLOS authors have the option to publish the peer review history of their article (what does this mean?). If published, this will include your full peer review and any attached files.

Reviewer #1: Yes: Robert Jandl

Reviewer #3: No

Reviewer #4: No

---

## [Author Response · Author response to Decision Letter 1]

10 Jun 2020

6. Review Comments to the Author

Reviewer #1

Reviewer #1: I want to congratulate the authors to an exceptional manuscript. The text has a high potential to be used in classrooms. The coverage of the topic is comprehensive. A particular value is the availability of the code of the data evaluation. The description of the data has the same high quality as the main text itself. I am greatly impressed by the quality of the submission and hope that it will find wide recognition.

Authors: Thank you for your valuable and precise review work.

 

Reviewer #3 

Reviewer #3: This is an interesting analysis of a 20 year's time series of measured element concentrations in forests soil solution in relation to different environmental variables. The work adds to current knowledge on progressive acidification of forest soils intensified by atmospheric nitrogen and sulphur deposition. The authors elaborated further on a first version of their manuscript taking into account various comments of two reviewers. I think this has substantially improved the overall quality of the paper, but there are still some issues that need to be solved before it can be published, particularly concerning the modeling analysis.

Authors: Thank you for your valuable questions and suggestions.

detailed comments

Reviewer #3: L18: ‘Acidification indicators remained stable at high levels...’ : I think this terminology is too vague for an abstract. Could you please formulate it in more concrete terms?

Authors: The sentence was now changed into "... In strongly acidified soils (soil pH below 4.2), acidification indicators changed only slowly"

Reviewer #3: L23 (and further): The term ‘N leaching’ is used throughout the paper, which I think is not entirely correct, since dissolved organic forms of N (DON) are not included. I suggest to consistently write ‘nitrate leaching’ (assuming that ammonium leaching is negligible).

Authors: You are right, we changed now "N leaching" to "nitrate leaching" throughout the document.

Reviewer #3: L25-26: I suggest to drop the following sentence: ‘Therefore, we suggest a restricted use of N leaching as an eutrophication indicator’. Plenty of studies have shown that NO3 leaching is an important indicator of eutrophication/N saturation, but since forests are natural ecosystems with a huge variety in environmental conditions there logically also must be exceptions.

Authors: We removed this sentence and changed the wording in the preceding one.

Reviewer #3: L58-59: as the BC/Al ratio is defined here, I think you should also mention what you mean with base cations (Ca2+ + K+ + Mg2+) in this sentence.

Authors: We inserted the names of the ions included for base cations calculation. Is this what you asked for?

Reviewer #3: L69-71: I suggest to refer to the groundwater directive here. DIRECTIVE 2006/118/EC OF THE EUROPEAN PARLIAMENT AND OF THE COUNCIL of 12 December 2006 on the protection of groundwater against pollution and deterioration (https://eur-lex.europa.eu/LexUriServ/LexUriServ.do?uri=OJ:L:2006:372:0019:0031:EN:PDF).

Authors: Switzerland is not in the EU so reference was now made to the Swiss legislation.

Reviewer #3: L137 and L178: it is not clear to me whether and how the alkalinity data were actually used in your analysis. I assume ANC was calculated rather as the balance between anions and cations? If alkalinity itself was not used, the corresponding part of the sentence in L137 should be removed.

Authors: Titrated alkalinity (not ANC) was part of the quality control for ion balance and calculated conductivity so it should stay in. 

Reviewer #3: L224 and Fig. 2: a quadratic term was used in order to allow modeling of a non-linear relationship. This approach has not been described in the materials and methods section. I actually wonder if a linear mixed additive model (gamm) applied on the untransformed data would not be more appropriate here, as it would probably be more easy to understand. The current graphs are somewhat misleading: it looks as if the curve flattens of at higher BS, but instead it is the opposite: if the Y-axis would be in regular units instead of log-scale, one would see an exponentially increasing curve.

Authors: A log transformation is quite common for concentration data. Thus, we cannot see the point that we should try to replace the transformation of the dependent variable with a more sophisticated model of the independent variables. Actually, after recalculation the model with base saturation loses the polynomial term. 

For answering the question on gamm, we compared two models with BC/Al as dependent variable and pH(CaCl2) as independent variable (Figure 2 b). 

The first model uses BC/Al after log transformation and a linear model with the independent variable pH as polynom with 2 degrees of freedom. 

model1 <- lm(LBCAL~poly(PHCACL,degree=2),data_BC_AL_ratio1)

The second model uses BC/Al without transformation and a gam model with the independent variable pH as polynom with 3 degrees of freedom.

model2 <- gam(BCAL ~ s(PHCACL,k=3),data=data_BC_AL_ratio1)

The residual plots show a good normal distribution of the residuals in model 1 and an S shaped distribution in model 2. The Tukey-Anscomb plot for model 1 show a rather homogenous distribution of the points whereas model 2 is clearly heteroscedastic. Both problems can be solved with a log transformation of the dependent variable. The R2 is somewhat lower in model 2 (R2 adj=0.527) than in model 1 (R2 adj=0.547), the AIC much higher (722.8 vs. 229.3).

Reviewer #3: L226: I do not understand this. BC/Al ratio at the start of the 5 year period was included as a predictor for the BC/Al ratio?

Authors: The hypothesis tested was that the change of BC/Al depends on the degree of acidification. The initial BC/Al ratio was taken as a measure for this. An introductory sentence was inserted.

Reviewer #3: Section 3.1.3 and 3.1.4: I have some doubts on the modeling analysis. The number of independent variables in the initial models (about 10) is actually quite large for the number of sites (n = 47). While there is no general rule of restriction, a large number of variables could easily lead to overfitting of the model and thus wrong interpretation and results. I am not entirely convinced that the applied backward selection with AIC/BIC excluded the least important variables. Instead, it might have been a better approach to restrict the initial number of variables (pre-selection) based on present knowledge or to do a forward selection.

Authors: In this regression analysis, the initial BC/Al ratio and solution pH have a higher degree of freedom as they vary within 5 year intervals. Base saturation is the predictor with the lowest degrees of freedom. It varies at the level of depth*plot and has 89 observations. 

Reviewer #3: Table 2 and 3: The data for the dependent variables BC/Al ratio and N leaching were log-transformed, as mentioned in the caption of both tables. Log-transformation changes the relationship between the dependent and independent variables and may therefore lead to a wrong interpretation of the results. This choice should thus be clearly motivated, which information is currently missing in the materials and methods section.

Authors: Models with and without log transformation were compared and the distribution of the residuals checked for normality. We have added a sentence to the statistics section. However, for concentration data log transformation is very common. 

Reviewer #3: L253-254: you wrote that normality and homogeneity of variance of the residuals was checked, but it is necessary to show also graphs of normalised residuals vs. fitted values and of normalised residuals against each independent variable in the model, in order to allow the reader to judge the validity of models. In L258-259 it is stated that this information is included in the Supplementary information, but I couldn’t find it there.

Authors: This information has been included as part of the R documentation which is available in the dryad repository (https://datadryad.org/stash/share/PjiHqj7uwJK13MBFbbbiRVeoO_DI1byCEPpHDD3bDms). No change has been made.

 

Reviewer #4: 

Reviewer #4: This manuscript assesses recent soil acidification in Swiss forest stands in response to atmospheric acidic deposition using soil solution chemistry measured over the last 20 years. This is a valuable dataset, due to the large number of plots that have been monitored for many years in different soil types across Switzerland in two major forest types (spruce and beech) and exposed to different levels of atmospheric deposition.

I was not reviewer of the first version of the manuscript. It seems that the authors have made a great deal of effort in improving the quality of the manuscript. However, I have major concerns about the quality of the revised manuscript: 1) there is no research questions and hypothesis, 2) the description of the current state of knowledge is thin and the authors’ results are not integrated in a broader context, 3) the way of presenting some results is questionable. I explain the three points:

Authors: Thank you for your review work.

Reviewer #4: 1) Missing research question: the authors do not ask any research question and do not postulate any hypotheses. It is therefore difficult to assess whether their statistical analyses allow them to answer their questions! In recent decades, air pollution reduction policies have resulted in a large reduction in sulphur emissions and, to a lesser extent, in nitrogen emissions. However, the authors do not question the effect of these measures on soil acidification in their introduction. It is not clear whether the focus is on trends of soil acidity indicators or on the effects of forest health.

Authors: We have now addressed research questions and discussed the air pollution reduction more in detail in the introduction.

Lines 41-46:

due to mitigation measures, the deposition of acidifying substances in Europe, in particular of sulfur compounds, has decreased in recent years. In consequence, sulfate concentrations in soil solution have decreased significantly ([6]). However, the development of nitrogen indicators is more divergent ([7]). In Canada, for example, the chemical recovery of streams was slower than expected due to the reduction of acid deposition ([8]).

Lines 109-122:

The aim of the present study is to analyze trends in soil solution data collected over a period of 20 years from currently 47 plots of the Intercantonal Forest Monitoring Program in Switzerland ([41]). The observed changes in the element concentration of the soil solution measurements were analyzed with respect to international critical limits and other threshold values in order to assess the risk of acidification and eutrophication effects on forest health in Switzerland. The following research questions were addressed:

I. Are critical limits and thresholds exceeded?

II. Do the reductions in acid depositions translate into corresponding changes in soil solution chemistry?

III. What are suitable predictors for the risk of high nitrate leaching?

The parameters measured in this monitoring program are based on the Guidelines on Reporting Monitoring and Modelling of Air Pollution Effects of the Geneva Air Convention ([42]). 

Reviewer #4: 2) Insufficient description of the current state of knowledge: There are few references to recent studies related to the trends of the soil solution chemistry in response to declining acidic deposition. Yet many publications have been published in the last decade about the potential recovery of soil solution in forest soils and of surface waters in Europe and North America (in particular in countries participating in ICP-Forests and ICP-Waters) in response to declining acidic deposition. In addition, a major European study has been recently published (Johnsson et al. 2018, Global Change Biology, DOI: 10.1111/gcb.14156) on the response of soil solution chemistry in European forests. There was no mention of this study in the revised manuscript. Since the current state of the knowledge is insufficiently described, the introduction does not tell which knowledge gap this manuscript aims to fill. Also in the discussion, the results are not integrated in a larger context, not even at the national scale. There are few references to Swiss and European studies related to soil (solution) acidification.

Authors: Thank you. The results from ICP Forests are now introduced. We made also reference to the Canadian study.

Reviewer #4: 3) Several concerns about the way of presenting results:

3a. I could not see a systematic way in presenting the data. The level of data aggregation for the different graphs and tables seems to be different, which makes the manuscript heterogeneous and difficult to understand (e.g. analysis of different depth intervals in figures). The tables are also difficult to read because units are not included. For instance, it would help the reader to know which year corresponds to the intercept and the units of intercept and changes (relative, absolute slope?) in the tables presenting the linear models.

Authors: The level of data aggregation depends on the type of data so this cannot be unified. But we have now introduced units into the tables where appropriate. The unit of the intercept has no significance for the interpretation of the results except when all predictors are set to zero. The main reason for the regressions against time was to get estimates for the single years which are corrected for the changing sample size.

Reviewer #4: 3b. The topic of the manuscript is soil solution chemistry over the last 20 years, but I have not learnt much about the quality of the soil solution and how it has changed over time in the studied plots. A statistical summary of concentrations of individual cations and anions and the corresponding trends would have been useful. To present only the BC/Al ratio is insufficient to assess changes in soil solution chemistry. Knowing the behavior of sulphur, base cations and aluminum concentrations (or fluxes if available) is essential to better understand the effect of declining atmospheric deposition. The authors refer to a report illustrating time series at each plot. The link in the reference list is, however, not valid. After searching in Internet, I found this interesting report. The graphs show that the temporal trends differ between the plots and depths and, interestingly, that acidity indicators are stabilizing in the last years in some plots after a decrease in the first years of observation.

Authors: We have corrected the link to the internal report. A new table has been introduced with mean concentrations per depth class and time trends.

Reviewer #4: 3c. Table 2: this analysis is not convincing. The authors mean in this graph that even a low percentile of measurements below the critical thresholds at a given plot would have a critical impact, which has not been reported in the literature. A more detailed analysis including information about the distribution of BC/Al values would be more informative.

Authors: The CCE documentation does not clarify if the critical thresholds are to be exceeded as annual average or as single value. We decided to use the latter as a root will not see the average but an actual value. The use of the percentile was only to avoid single outliers and make thus the analysis more robust. But we agree that a density distribution would be helpful. We added this to the Supplementary Material..

Reviewer #4: 3d. Figure 2: The calculation of BC/Al ratio at pH > 6.5 is misleading. The aluminum concentrations are very low, close to the detection limit and therefore cannot be quantified precisely. Very small differences in Al concentrations lead to large variations in BC/Al. In that respect, the quantification limit of Al should be reported to assess the uncertainty related to high BC/Al. In addition, inorganic Al was measured as difference before and after passing the samples through an ion exchanger and therefore uncertainties add up. Another tricky point in this figure is the comparison of soil and soil solution parameters using large depth intervals (<70 cm and >70 cm). The authors do not explain how BC/Al, BS and pH measurements were aggregated in this large depth interval. In this analysis, the presence of calcareous parent material in the subsoil might explain the discrepancies between BS (Fig. 2B) and soil pH (Fig. 2C) (see Blaser et al. 2008, https://doi.org/10.1002/jpln.200625213). Acidified forest soils on calcareous parent material in Switzerland usually have a strong base saturation gradient with depth. Also soil water regime (hydromorphy) and reduced drainage play an important role in the chemistry of these soils.

Authors: The comparison between soil solution and solid phase was based on the chemical criterial of the solid phase at the depth of the lysimeter. This has been already stated in the Methods section but an additional remark has been made now. 

Lines 162-163:

Actual depths vary according to soil condition but a frequent sampling design was 20, 50 and 80 cm.

The large depth intervals were only used as an additional binary indicator of soil depth as a finer discrimination was not significant. The discussion on the presence of calcareous parent material has therefore no relevance for the presented results.

It is correct that the BC/Al ratio at pH>6.5 is subjected to a large error. However, when these data would be omitted the data set would be subjected to a large bias. It is common use to set the detection limit for data analysis when working with concentration data with part of the samples below the detection limit.

Reviewer #4: 3e. Figure 3 is very interesting, because few studies have assessed Ca weathering rates and compare them to Ca leaching. However, information about temporal trends is missing. It is known from many studies that concentrations of base cations in soil solution have decreased in the last decade. The driver responsible for this decrease is not completely clear (chemical equilibrium due to declining sulphate concentrations in soil solution? declining atmospheric deposition of base cations? increase in root uptake? ). The temporal analysis of BC leaching from the soil is therefore also relevant because the rate of BC loss from the soil might have slowed down in the last decade, which is still a relevant information to assess the effect of air pollution reduction measures.

Authors: We tried to include two time periods into Figure 3 but this gets then very confusing. We therefore decided to add a sentence explaining how much the results are changed when only later data are included.

Lines 347-350:

The same analysis with leaching rates at different time periods slightly reduced the proportion of plots with a negative balance. For instance, between 2015-2018, Ca leaching exceeded the Ca weathering input in 83% of the plots.

 The differences are actually very small as for almost all the plots the exceedance is remaining.

Reviewer #4: 3f. Figure 5 is interesting. Fig. 5A shows that high BC/Al ratios at a given time are likely to decrease in the following years, while low BC/Al ratios are likely to stabilize or even increase. In lign 312, the authors write “The decreasing trends were getting weaker the stronger the soils are acidified”. I do not understand why the predicted values in 5B and 5C actually show the opposite trend.

The predicted values in Fig. 5 are the result of a multivariate analysis. Fig. 5B and C show the result of changes in base saturation and pH when BC/Al is kept constant (at the mean value). They show the relevance of buffer ranges: at higher pH and/or base saturation soils are in the CaCO3 buffer range where changes have a positive sign. 

Other specific comments:

Reviewer #4: • l. 17: remove “in Swiss forest soils”, not necessary

Authors: o.k., removed.

Reviewer #4: • l.18. “remained stable at high level”: formulation is not clear

Authors: The sentence has been changed

Lines 17-20:

In strongly acidified soils (soil pH below 4.2), acidification indicators changed only slowly, possibly due to high buffering capacity of the aluminum buffer (pH 4.2 – 3.8).

Reviewer #4: • l. 20 “an increasing acidification”: not clear. An increasing acidification rate? Acidification is happening anyway.

Authors: O.k., we introduced "acidification rate".

Lines 20-22:

In contrast, in less acidified sites we observed an increasing acidification rate, reflected, for example, by the continuous decrease in the ratio of base cations to aluminum (BC/Al ratio).

Reviewer #4: • l. 28-29: “Taken together, this study provides evidence of increasing soil acidification in Swiss forest stands.” This is not clear. Do the author mean that the rates of acidification are increasing? Fig. 5A illustrates a more balanced view.

Authors: We changed "increased" by "anthropogenic".

Lines 30-31:

Taken together, this study provides evidence of anthropogenic soil acidification in Swiss forest stands.

Reviewer #4: • l.31: “climate change”: meteorological parameters were used in the analysis. Climate change was not analysed.

Authors: We replaced "climate change" by "drought" as this was actually analysed.

Lines 31-33:

The underlying long-term measurements of soil solution provides important information on nutrient leaching losses and their dependence on drought.

Reviewer #4: • l.45-48: The link between the exceedance of critical loads for acidity and the decreasing BC/Al ratio in Pannatier et al. (2011) is not clear. Declining BC/Al does not mean that critical loads are exceeded.

Authors: We changed the sequence of the corresponding sentence. 

Lines 54-59:

For instance, Graf Pannatier et al. ([14]) observed low BC/Al ratios in the topsoil in two out of five Swiss long-term forest monitoring sites, i.e. an exceedance of the critical loads for acidity. In addition, a decrease in the BC/Al ratio has been found in two out of five plots between 1999 and 2007, which can be interpreted as an ongoing acidification during this time period.

But there is no doubt that a BC/Al ratio <1 as shown in this study can be interpreted as an exceedance of the critical load for acidity as this was part of the definition.

Reviewer #4: • L.51: “is of crucial importance”: why?

Authors: "Crucial importance " was now replaced by "important part".

Lines 61-62:

Concerns about forest health led to the initiation of forest monitoring programs in the 1980s, where monitoring of soil solution is an important part ([15]).

Reviewer #4: • L. 132-133: suction cups? at which depth? Basic information on the type of suction cups used in this study and sampled depths should be available in this section.

Authors: A sentence has been introduced to explain the type and the depth of the soil solution samplers.

Lines 161-164:

For each site and soil depth, eight soil solution samplers (ceramic suction cups, 0653X01-B0.5M2, Soilmoisture Equipment Corp.) were installed in the topsoil and five in the subsoil. Actual depths vary according to soil condition but a frequent sampling design was 20, 50 and 80 cm.

Reviewer #4: • L. 159: what are the vegetation parameters?

Authors: It is not clear what is meant here but we added a remark that we used current vegetation cover for the hydrological model.

Lines 188-190:

The amount of leaching water in mm was calculated using the hydrological model Wasim-ETH ([44]) taking into account soil characteristics (pF curve, texture), current vegetation cover and daily meteorological data interpolated for each site ([34]).

Reviewer #4: • L. 211: reference in prep is not suitable

Authors: We changed this to "unpublished results"

Reviewer #4: • L. 289: Reference to buffer ranges is missing

Authors: It is not clear what is asked here: a reference to a paper or an indication of the pH ranges. We decided to do the latter and inserted a pH range for the exchange buffer range.

Reviewer #4: • L. 318, l. 470: not comma before that. Remove everywhere

Authors: o.k.

Reviewer #4: • L. 401: “The results are confirmed by findings of the long-term forest monitoring of ICP Forests in Switzerland ([66]).” This sentence is not correct since ref 66 is much older.

Authors: We replaced this reference by Graf Pannatier et al. 2011 which is more recent.

Reviewer #4: • L. 445-447: “Based on our results we question the reliability of the N concentration of the leaves as an indicator for eutrophication, since our measurements show that in beech leaves today they are no longer correlated with N deposition, which was the case in the 1980s”. Not shown by data from this manucript.

Authors: But a reference is given (Braun et al. 2020) therefore we think is well supported.

Reviewer #4: • L. 476-L. 475. “Soil acidification has negative consequences for forest health (…). This conclusion is not supported by the findings of this manuscript.

Authors: It is supported by findings from the the long-term Intercantonal Forest Observation Program. Detailed references are given therefore we think this is well supported.

Reviewer #4: • L. 479: no analysis of climate change in this manuscript. Meteorological parameters were used. The manuscript did not show that the droughts were related to climate change. References are needed to make this link.

 Authors: We replaced now "climate change" by "drought". It would lead too far to lead the discussion on the relation between climate change and drought at this place and would actually be not relevant for the conclusions.

7. PLOS authors have the option to publish the peer review history of their article (what does this mean?). If published, this will include your full peer review and any attached files.

Do you want your identity to be public for this peer review? For information about this choice, including consent withdrawal, please see our Privacy Policy.

Reviewer #1: Yes: Robert Jandl

Reviewer #3: No

Reviewer #4: No

---

## [Editor Report · Decision Letter 2]

23 Jun 2020

Soil solution in Swiss forest stands: a 20 year's time series

PONE-D-19-35297R2

Dear Dr. Braun,

We’re pleased to inform you that your manuscript has been judged scientifically suitable for publication and will be formally accepted for publication once it meets all outstanding technical requirements.

Kind regards,

Julian Aherne

Academic Editor

PLOS ONE

Additional Editor Comments (optional):

This is an important contribution. The authors have done an excellent job in addressing all reviewers’ concerns in the revised manuscript. I recommend that it is accepted for publication.
---

## [Editor Report · Acceptance letter]

29 Jun 2020

PONE-D-19-35297R2 

Soil solution in Swiss forest stands: a 20 year's time series 

Dear Dr. Braun:

I'm pleased to inform you that your manuscript has been deemed suitable for publication in PLOS ONE. Congratulations! Your manuscript is now with our production department. 

Kind regards, 

on behalf of

Dr. Julian Aherne 

Academic Editor

PLOS ONE